# Conformational trajectory of the HIV-1 fusion peptide during CD4-induced envelope opening

Bhishem Thakur[1], Revansiddha H. Katte [2], Wang Xu [2], Katarzyna Janowska[1], Salam Sammour[1], Rory Henderson [1,3], Maolin Lu[2], Peter D. Kwong [4,5] & Priyamvada Acharya [1,6,7] ✉

The hydrophobic fusion peptide (FP), a critical component of the HIV-1 entry machinery, is located at the N terminus of the envelope (Env) gp41 subunit. The receptor-binding gp120 subunit of Env forms a heterodimer with gp41. The gp120/gp41 heterodimer assembles into a homotrimer, in which FP is accessible for antibody binding. Env conformational changes or "opening" that follow receptor binding result in FP relocating to a newly formed inter-protomer pocket at the gp41-gp120 interface where it is sterically inaccessible to antibodies. The mechanistic steps connecting the entry-related transition of antibody accessible-to-inaccessible FP configurations remain unresolved. Here, using SOSIP-stabilized Env ectodomains, we visualize that the FP remains accessible for antibody binding despite substantial receptor-induced Env opening. We delineate stepwise Env opening from its closed state to a functional CD4-bound symmetrically open Env in which we show that FP was accessible for antibody binding. We define downstream re-organizations that lead to the formation of a gp120/gp41 cavity into which the FP buries to become inaccessible for antibody binding. These findings improve our understanding of HIV-1 entry and delineate the entry-related conformational trajectory of a key site of HIV vulnerability to neutralizing antibody.

The HIV-1 envelope glycoprotein (Env), a homotrimer of gp120-gp41 heterodimers, mediates virus entry into host cells[1–3]. The gp120 subunit engages host receptors, while the gp41 subunit contains a fusion peptide (FP) that inserts into the host membrane to effect host and virus membrane fusion[1,4–6]. Prior to its binding to host receptors, the HIV-1 Env is characterized by a closed configuration with gp120 protomers packed against each other and the gp41 subunit, while the highly conserved and immunodominant coreceptor-binding region at the Env trimer apex remains occluded by packing of the first and second (V1V2) as well as the third (V3) variable loops (Fig. 1A)[2,3,7,8]. At the trimer base, FP comprises a hydrophobic stretch of 15–20 amino acids at the gp41 N terminus[5,9]. FP is a site of vulnerability to broadly neutralizing antibodies (bnAbs) and thus is a focus of vaccine development efforts[7,10,11].

HIV-1 Env uses its gp120 subunits to engage the CD4 receptor on the surface of human immune cells. CD4-induced conformational changes have been structurally characterized in virus-associated Env by cryo-electron tomography (cryo-ET)[6,12,13], while high-resolution

[1]Duke Human Vaccine Institute, Duke University, Durham, NC, USA. [2]Department of Cellular and Molecular Biology, School of Medicine, University of Texas at Tyler Health Science Center, Tyler, Texas, USA. [3]Department of Medicine, Duke University, Durham, NC, USA. [4]Aaron Diamond AIDS Research Center, Columbia University Vagelos College of Physicians and Surgeons, and Department of Biochemistry and Molecular Biophysics, Columbia University, New York, NY, USA. [5]Vaccine Research Center, National Institute of Allergy and Infectious Diseases, National Institutes of Health, Bethesda, MD, USA. [6]Department of Surgery, Duke University, Durham, NC, USA. [7]Department of Biochemistry, Duke University, Durham, NC, USA. ✉e-mail: priyamvada.acharya@duke.edu

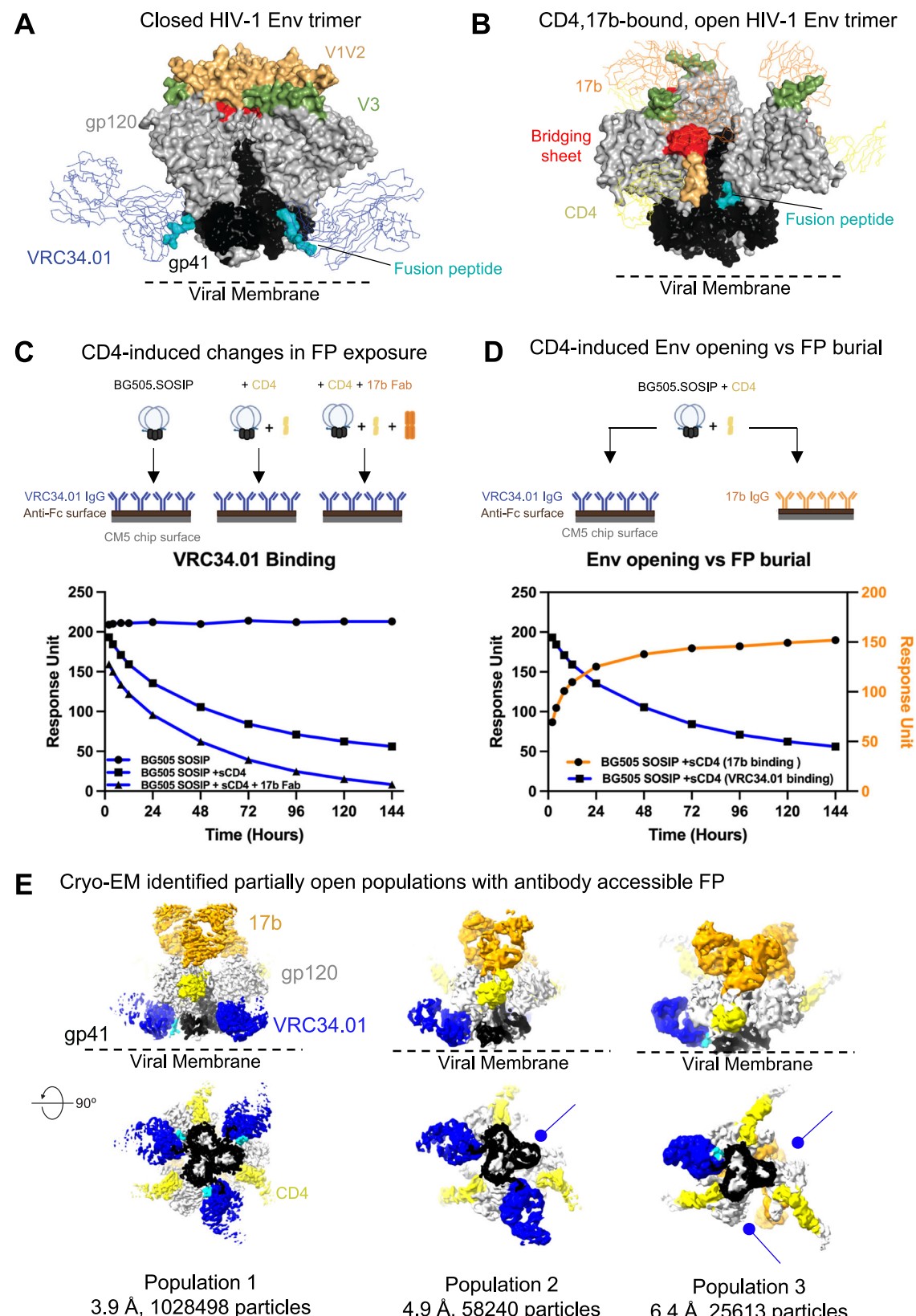

**A** Closed HIV-1 Env trimer

**B** CD4,17b-bound, open HIV-1 Env trimer

**C** CD4-induced changes in FP exposure

**VRC34.01 Binding**

**D** CD4-induced Env opening vs FP burial

**Env opening vs FP burial**

**E** Cryo-EM identified partially open populations with antibody accessible FP

Population 1
3.9 Å, 1028498 particles

Population 2
4.9 Å, 58240 particles

Population 3
6.4 Å, 25613 particles

structural definition of receptor-induced Env opening has been obtained by single-particle cryo-EM analysis of stabilized, soluble Env ectodomains[14–16]. Both lines of evidence have synergized to facilitate our understanding of HIV-1 entry-related intermediate states and have enabled visualization of functionally relevant Env structural changes across resolution scales.

CD4-induced Env conformational changes, collectively termed as "Env opening", include rigid-body displacement and rotation of the gp120 subunits resulting in up to ~40 Å shift in the positioning of the V1V2 base (Fig. 1B)[17]. Env opening is accompanied by internal rearrangements within gp120 that involve disruption of inter-protomer interactions formed by the gp120 V1V2 and V3 regions, release of the V3

**Fig. 1 | Time-dependent conformational changes in HIV-1 BG505 SOSIP Env upon incubation with CD4. A** Structure of pre-fusion, pre-receptor, closed HIV-1 Env (PDB: 5I8H) bound to broadly neutralizing, fusion peptide-directed antibody VRC34.01. The Env is shown in surface representation with the gp120 subunits colored light gray, and within the gp120 subunits, the V1V2 loop colored wheat, V3 loop olive and the residues contributing to the bridging sheet in the open Env colored red. The gp41 subunits are colored black with the fusion peptide (FP) colored cyan. The antibody VRC34.01 is shown in ribbon representation bound to its FP-centered epitope. **B** Structure of pre-fusion, CD4-bound open HIV-1 Env bound to CD4-induced antibody 17b. The Env is colored similarly as in panel A. CD4 is shown as a yellow ribbon and 17b Fab is shown as an orange ribbon. **C** Surface plasmon based binding (SPR) analysis monitoring FP burial. Env was incubated at 25 °C with either sCD4 alone or with CD4 and the coreceptor mimicking antibody 17b. At different time-points after incubation, binding was measured to the fusion peptide targeting antibody VRC34.01. Some elements of the SPR schematic were created in BioRender. Acharya, P. (2025) https://BioRender.com/m74zcsz.
**D** Simultaneous Env opening and fusion peptide burial were measured by incubating Env with CD4 and at different time-points injecting over a VRC34.01 IgG or a 17b IgG surface. Source data for panels (**C, D**) are provided as a Source Data file. Data shown are representative of at least two independent experiments. **E** Cryo-EM reconstructions of three distinct populations of CD4/17b-bound, partially open Env bound to VRC34.01. Population 1 is bound to VRC34.01 at all three sites. The blue arrows indicate sites unoccupied by VRC34.01 in Populations 2 and 3.Source Data.

loop, and formation and exposure of the bridging sheet. The V3 loop and bridging sheet are the structural elements that form the binding site for a GPCR coreceptor, either CCR5 or CXCR4[18–20]. The structural signatures of CD4-induced Env opening include the bridging sheet and the α0 helix in gp120 that were first defined in crystal structures of CD4-bound monomeric gp120[4]. CD4-induction of Env also re-organizes the gp41 subunit[17,21] resulting in burial of FP within a gp41 cavity such that it is no longer accessible for antibody binding (Fig. 1A, B).

While high-resolution structural details have been elucidated for FP in an antibody-accessible conformation (the closed configuration of Env prior to receptor engagement)[7] and in an antibody-inaccessible conformation after CD4 receptor-induced opening of Env[17], the mechanistic details of this FP relocation remain unclear. Here, we use conformation-sensitive antibodies as molecular probes to simultaneously track the trajectories of Env opening and of FP accessibility. For FP accessibility, we used the prototype FP-directed antibody VRC34.01[7], isolated from a chronically HIV-1-infected individual, which binds at an epitope comprised primarily of the gp41 FP residues 512–519 (contributing ~55% of total interactive surface area) and gp120 glycan N88 (~26% of the total interactive surface area). For Env opening, we used the CD4-induced (CD4i) antibody 17b to assess the formation and exposure of the bridging sheet upon CD4-triggering of Env. As the formation of the bridging sheet requires disruption of the V1V2 cap at the trimer apex and at least partial Env opening, binding to 17b was also an indicator of Env opening[4,22].

Here, using Env ectodomains stabilized by an intraprotomer gp120/gp41 disulfide and an Ile to Pro change in gp41 (SOSIP)[8], we perform cryogenic electron microscopy (cryo-EM) to define intermediates where FP remains accessible to antibody binding despite substantial Env opening. Among these conformations are populations with their gp120 protomers either partially rotated from the pre-receptor closed Env conformation or more substantially rotated to resemble the geometry observed in the CD4-induced fully open conformation described previously[17,23]. The partially rotated gp120 is associated with antibody-accessible FP, whereas further gp120 displacement along an axis orthogonal to the central trimer axis resulted in FP burial, suggesting an association of FP burial with the extent of gp120 displacement. Taken together, our data provide evidence that accessibility of FP to antibody binding persists post-receptor engagement despite substantial Env opening. Furthermore, we define the mechanistic steps that lead to FP burial and antibody inaccessibility upon further Env opening. Our results resolve several gaps in our knowledge of HIV-1 entry and provide information relevant to the development of vaccines and therapeutics.

## Results

### FP remains accessible for antibody binding after CD4-induced Env opening

To assess CD4-induced changes in FP accessibility, we measured binding to the FP-targeted antibody VRC34.01 at different time points following incubation of BG505.SOSIP Env with either CD4 alone, or together with the fragment antigen binding (Fab) of the coreceptor-mimicking antibody 17b that recognizes an epitope presented upon CD4-induced Env opening (Fig. 1C and Supplementary Fig. S1). VRC34.01 binding decreased after CD4-induction, and the decrease was more profound in the presence of 17b Fab. A control experiment without the addition of CD4 or 17b showed no change in VRC34.01 binding to BG505.SOSIP Env. We next assessed simultaneous changes in FP exposure measured by binding to VRC34.01, and Env opening measured by binding to 17b (Fig. 1D and Supplementary Fig. S1). 17b binding increased post CD4 addition, indicating Env opening and exposure of the bridging sheet, while VRC34.01 binding decreased.

To visualize the impact of CD4-bound Env conformations on FP positioning, we incubated BG505 SOSIP Env with CD4 and 17b Fab at 25 °C, and performed single particle cryo-EM on the Env complexes at selected time-points, 1.3 h (hr), 20 h, and 3 days, post CD4/17b addition, with VRC34.01 Fab added 30 min before the samples were vitrified for cryo-EM analysis (Figs. 1E, 2, and Supplementary figs. S2–S7, Table S1). We identified three particle populations across the three cryo-EM datasets that differed in their stoichiometries of bound VRC34.01 Fab (Fig. 1E, Supplementary Fig. S7, and Supplementary Table S2). Population 1 dominated at all three time points and consisted of a partially open Env in which each of the three gp120-gp41 protomers were bound to CD4, 17b Fab and VRC34.01 Fab. Another population, named Population 2, was detected at all three time points, albeit in smaller proportions relative to Population 1 (Supplementary Fig. S7). In Population 2, each of the three gp120 subunits were bound to CD4 and 17b Fab, while only two protomers were bound to VRC34.01 Fab. The proportion of Population 2 relative to Population 1 increased with longer incubation times. At the 3-day time point, a third population, named Population 3, was detected that resembled Populations 1 and 2 in their bound CD4 and 17b stoichiometries but only had a single VRC34.01 Fab bound, leaving two protomers not bound to VRC34.01.

While the experiments described above were performed by incubating Env with the ligands at 25 °C, similar trends were observed in SPR binding assays when the incubations were performed at 37 °C (Supplementary Figs. S1 and S9). A cryo-EM dataset obtained by incubating BG505 SOSIP with CD4 and 17b Fab at 37 °C for 2 h, followed by a 30 min incubation with VRC34.01 Fab at 37 °C prior to plunge freezing yielded structures representing Populations 1, 2 and 3. The appearance of the Population 3 structure earlier than could be detected in the 25 °C cryo-EM datasets suggested that the CD4-induced Env conformational changes are more rapid at 37 °C than at 25 °C. The recurrence of these structures across independent experiments highlights the reproducibility of these structural states.

In summary, we identified three populations of CD4-induced Env in our cryo-EM datasets with differing stoichiometries of bound VRC34.01. These results confirmed that FP remained accessible to VRC34.01 binding despite substantial Env opening and suggested FP accessibility to antibody binding is hindered at the sites in Populations 2 and 3 that were not bound by VRC34.01.

**A** Population 1 bound to CD4, 17b, and FP-directed antibody VRC34.01

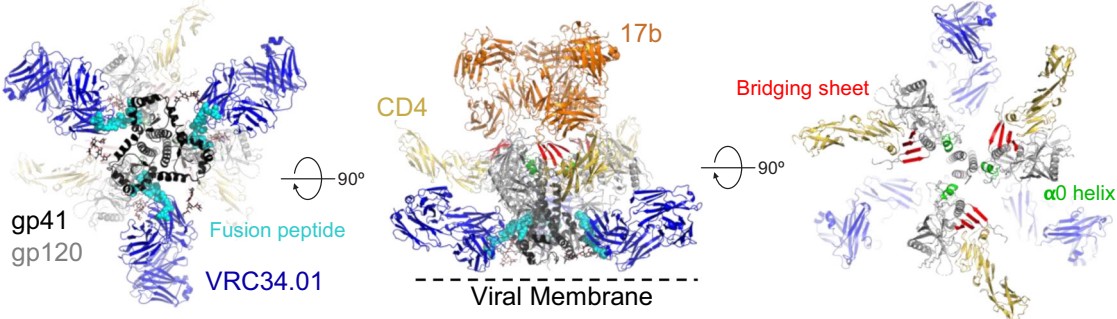

**B** Population 1 fit into *in situ* cryo-ET of CD4-bound Env intermediate

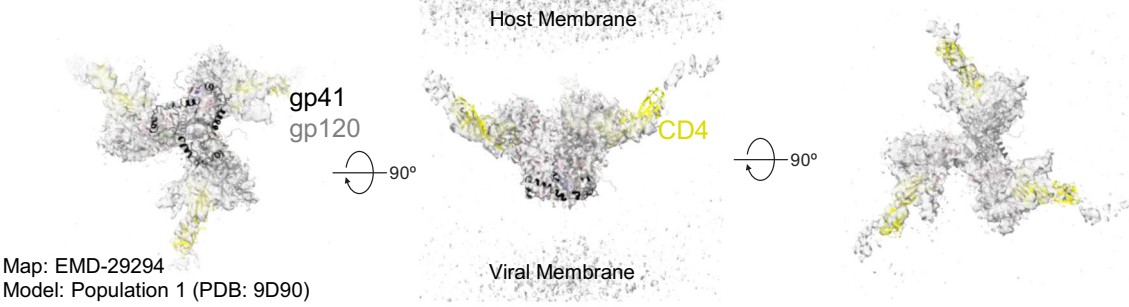

Map: EMD-29294
Model: Population 1 (PDB: 9D90)

**C** Population 1 comparison to Env in diverse conformations

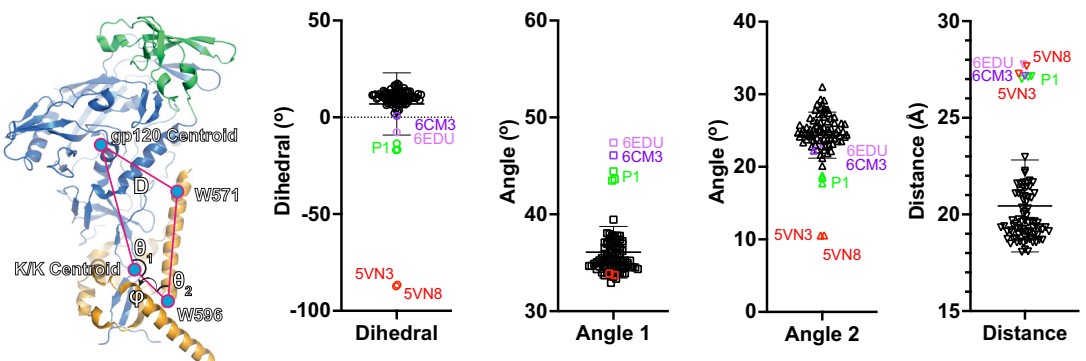

**D** Two distinct intermediate configurations of the HIV-1 Env FP

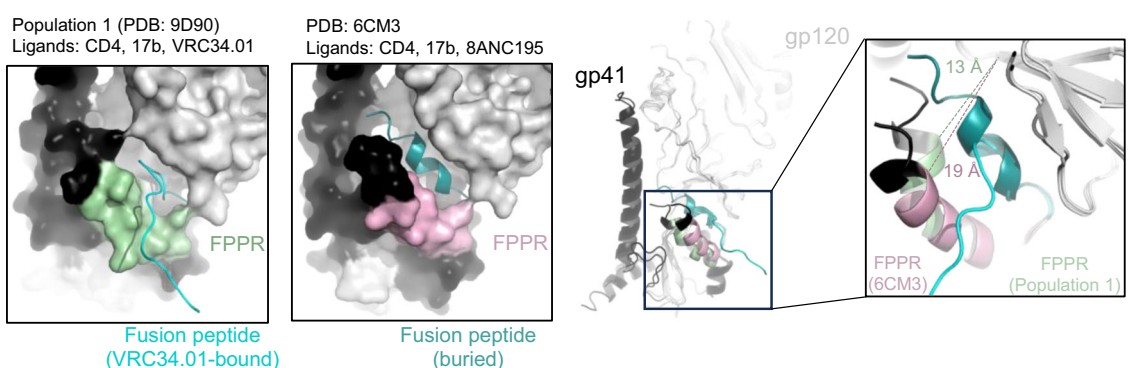

**A functional CD4-triggered partially open Env intermediate has an antibody-accessible FP**

Two distinct structural configurations of the FP have thus far been defined in the literature, one that is antibody accessible in the prefusion closed Env[7] and a second that is sequestered within a gp41-

gp120 pocket in a partially or fully open CD4-induced Env[17,21]. In this study, we have discovered new CD4-induced Env intermediates that are open enough to bind the bridging sheet-directed antibody 17b and yet retain the ability to bind a FP-targeting antibody. To understand Env-structural changes that enable CD4-induced opening, while the FP

**Fig. 2 | A partially open intermediate on the HIV-1 entry pathway retains FP accessibility to antibody binding. A** Three views of Population 1 structure shown in cartoon representation with gp41 colored black, gp120 gray, CD4 yellow, VRC34.01 blue, 17b orange. Glycans are shown as sticks. FP within the gp41 subunit is colored cyan. Within gp120, bridging sheet is colored red and **α**0 helix green. **B** Population 1 coordinates including Env (gp120 in gray, gp41 in black) and CD4 (yellow) fitted into the in situ cryo-ET reconstruction of a partially open CD4-bound Env (EMD-29294). **C** (Left to right) Vectors describing position of gp120 relative to gp41. The gp120 structure (blue), gp120 V1/V2 region (green), and gp41 (orange) in the closed state overlayed with the centroid locations depicting the dihedral, angles, and distances describing the position of gp120 relative to gp41. Dihedral, angle, and distance values for closed, intermediate, and open state structures. Data shown as scatter dot plots with horizontal lines indicating the mean and standard deviation. List of structures used for the calculations is provided in Supplementary

Table 3. Source data are provided as a Source Data file. **D** (Left) Population 1 protomer shown in surface representation zoomed-in at the location of the FP. FP is shown in cartoon representation. The gp120 subunit is colored light gray, gp41 black, FP cyan and FPPR pale green. (Middle) One protomer of the partially open Env bound to CD4, 17b Fab and 8ANC195 Fab (PDB ID: 6CM3) shown in surface representation zoomed-in at the location of the FP (shown in cartoon representation). The gp120 subunit is colored gray, gp41 black, FP dark teal and FPPR light pink. (Right) Overlay of a Population 1 protomer with a protomer of a partially open CD4,17b,8ANC195-bound Env (PDB ID: 6CM3). The gp120 subunits were used for the superposition. Inset zooms in on the FP and FPPR. Zoomed-in panel is slightly rotated compared to zoomed-out view for better visualization. The solid lines (pale green for Population 1 and light pink for 6CM3) show the distance between FPPR residues Gln 540 and gp120 residue Phe 223.Source Data.

remains in an antibody-accessible configuration, we first examined Population 1, which was the dominant population in all the cryo-EM datasets (Figs. 1E, 2, and Supplementary Fig. S7, Tables S1 and S2). We selected the Population 1 reconstruction from the 1.3-hr time-point for our analysis as it contained the largest number of particles and the highest resolution among the Population 1 structures from the three datasets.

In Population 1, the gp120 subunits exhibited known structural markers of the CD4-induced conformation[2,4,17], including the bridging sheet at the 17b-binding interface and residues 63–73 assembled into the **α**0 helix (Fig. 2A and Table S2). The gp41 subunit appeared conformationally less perturbed and was bound to VRC34.01 with a similar interaction interface dominated by the FP and the gp120 N88 glycan, as previously observed in the structure of VRC34.01 in complex with the closed BG505 SOSIP (Fig. S10)[7]. Although no symmetry had been applied during the cryo-EM data processing, the three protomers were highly similar in the symmetrically open Population 1 intermediate (Fig. S10). Our Population 1 structure revealed a similar gp120 opening geometry as the cryo-ET structure of membrane-associated HIV-1$_{ADA.CM}$ Env bound to three membrane-associated CD4 molecules (Fig. 2B)[6], suggesting that Population 1 represents a physiologically relevant entry intermediate.

We next studied the Population 1 structure using a previously defined set of vectors that report on structural rearrangements associated with rigid body movements in gp120 relative to gp41 (Fig. 2C and Table S3)[24]. These vectors describe the orientation of gp120 relative to the gp41 three-helix bundle, capturing rotation of gp120 away from the trimer central axis and rotation orthogonal to the trimer central axis. These measures effectively capture differences between closed, open, and intermediate state Envs. All three Population 1 protomers clustered together in all measures examined and were similar to previously published structures of BG505 (PDB: 6CM3) or B41 (PDB: 6EDU) SOSIP bound to CD4, 17b and 8ANC195 Fab[21]. The Population 1 structures were distinct from previously published open and open occluded state structures (PDBs: 5VN3 and 5VN8, respectively) in gp120 rotations described by a dihedral angle ($\varphi$) that defines orthogonal rotation and angles ($\theta_1$ and $\theta_2$) describing rotations relative to the trimer central axis (Fig. 2C). However, the distance between the gp120 core and W571 was similar between the open and intermediate structures. Contrasting each with the closed state structure clusters indicates the open and open occluded structures occupy distinct angles in the dihedral and the angle between the gp41 three-helix bundle and gp120 termini, while the Population 1 and CD4, 17b, 8ANC195-bound Envs differ in the angle describing gp120 rotation away from the central trimer axis. In summary, our vector analysis indicates that the partially open Population 1 intermediate described here shifts the gp120 domains away from the central axis, while the open and open-occluded structures shift the gp120 domains orthogonal to the trimer central axis.

We compared the configuration of the FP in the previously published partially open CD4, 17b, 8ANC195-bound structure (PDB: 6CM3) and the partially open CD4,17b, VRC34.01-bound Population 1 structure resolved in this study (PDB: 9D90) (Fig. 2D). In the Population 1 structure (VRC34.01-bound), the FP was extended out of the Env core to bind the VRC34.01 antibody, whereas, in the 8ANC195-bound structures, the FP was buried in an intra-protomer gp120/gp41 hydrophobic pocket. The formation of the pocket for FP sequestration in the CD4, 17b, 8ANC195-bound structure was facilitated by a shift in the position of the FP proximal region (FPPR) primarily involving straightening of the FPPR helix creating space for FP burial. The distance between the Cα atoms of FPPR residue Gln540 and the gp120 residue Phe233 measured at 13 Å for the Population 1 structure and at 19 Å for the CD4,17b, 8ANC195 bound BG505 SOSIP structure (Fig. 2D). A similar FP configuration was observed in the B41-complex with CD4, 17b and 8ANC195, suggesting that this is an isolate-independent conformational state (Supplementary Fig. S11). The difference in FP accessibility while maintaining overall similar protomer geometry suggested that in this intermediate state the FP can either be antibody-accessible or it can be occluded. While VRC34.01 binds FP and stabilizes its accessible configuration, 8ANC195 stabilizes the FP occluded configuration.

In summary, we identified a CD4-triggered partially open Env intermediate on the HIV-1 entry pathway, with a protomer geometry that accommodates an antibody-accessible or a buried FP, with a conformational change in the FPPR being the major facilitator for this conformational switch of FP.

## Mechanism of downstream FP sequestering and antibody inaccessibility

In addition to the near symmetric, partially open Population 1 state where VRC34.01 was bound at each of the three FP sites, we also identified populations that were bound to either one or two VRC34.01 Fabs, leaving two and one FP sites unbound, respectively, despite saturating amounts of VRC34.01 Fab being used for sample preparation (Figs. 1E, and 3A, B and Supplementary Fig. S7 and Tables S1, S2). As expected, based on binding to antibody 17b, the bridging sheet and the α0 helix were formed in all gp120 protomers in the Population 2 and Population 3 structures (Fig. 3A, B, and Supplementary Table S2). Examining the unbound sites revealed FP sequestered within a gp120/gp41 pocket in an antibody-inaccessible configuration (Fig. 3C, D), thus providing a structural explanation for the lack of antibody binding to these FP sites.

Unlike the near-symmetric Population 1 structure, Populations 2 and 3 displayed marked asymmetries. To quantify Env opening, we measured interprotomer distances between the CD4-binding site gp120-residue Asp368 and gp120-residue Pro124 at the Env trimer apex (Fig. 3E, F). As previously recognized, the closed and open Env structures showed substantial differences in these distances[23] (Fig. 3E).

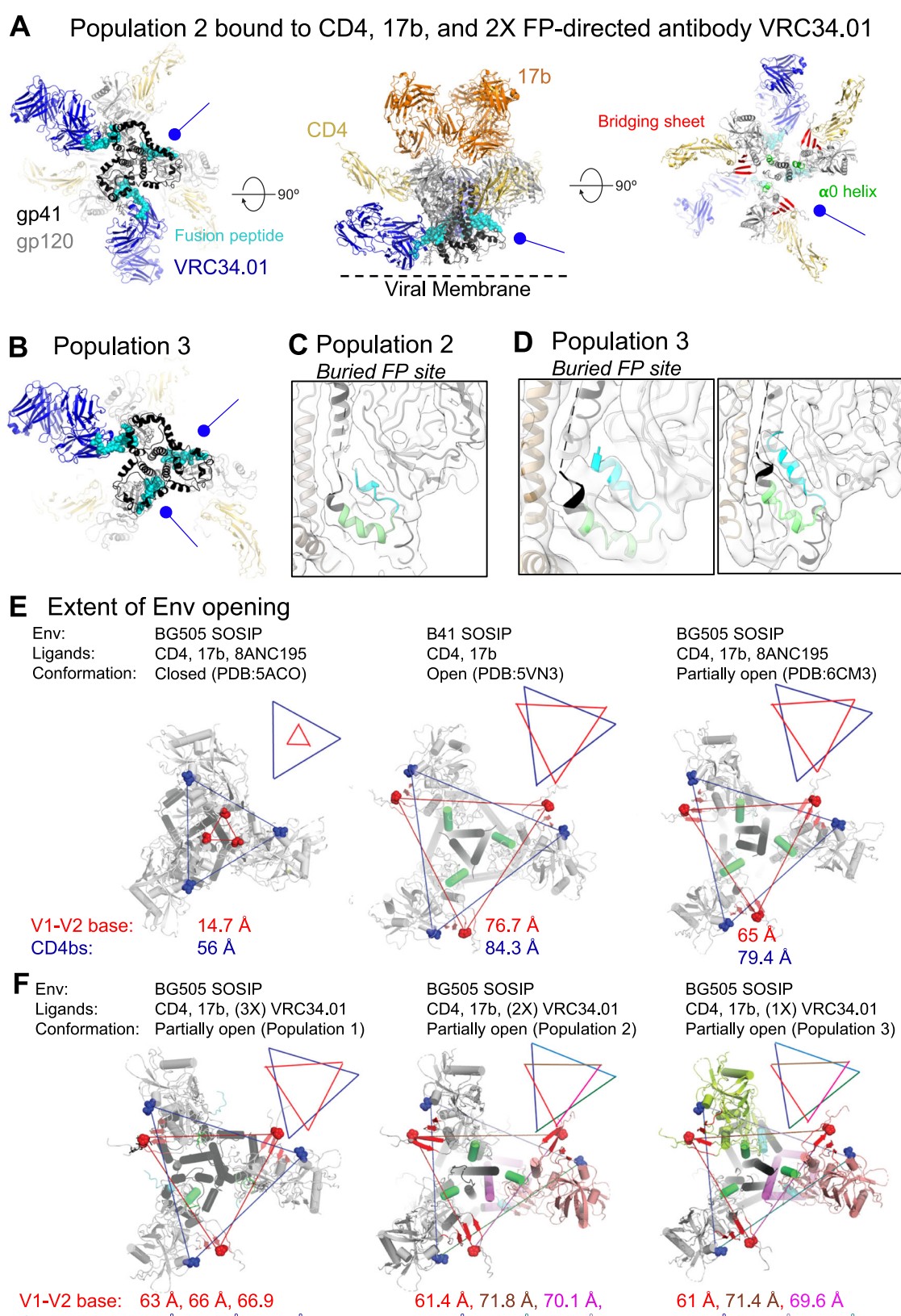

**A** Population 2 bound to CD4, 17b, and 2X FP-directed antibody VRC34.01

**B** Population 3

**C** Population 2
*Buried FP site*

**D** Population 3
*Buried FP site*

**E** Extent of Env opening

| Env: | BG505 SOSIP | B41 SOSIP | BG505 SOSIP |
| Ligands: | CD4, 17b, 8ANC195 | CD4, 17b | CD4, 17b, 8ANC195 |
| Conformation: | Closed (PDB:5ACO) | Open (PDB:5VN3) | Partially open (PDB:6CM3) |

V1-V2 base: 14.7 Å / 76.7 Å / 65 Å
CD4bs: 56 Å / 84.3 Å / 79.4 Å

**F**

| Env: | BG505 SOSIP | BG505 SOSIP | BG505 SOSIP |
| Ligands: | CD4, 17b, (3X) VRC34.01 | CD4, 17b, (2X) VRC34.01 | CD4, 17b, (1X) VRC34.01 |
| Conformation: | Partially open (Population 1) | Partially open (Population 2) | Partially open (Population 3) |

V1-V2 base: 63 Å, 66 Å, 66.9 / 61.4 Å, 71.8 Å, 70.1 Å, / 61 Å, 71.4 Å, 69.6 Å
CD4bs: 75 Å, 77.6 Å, 78.2 Å / 74.6 Å, 82.8 Å, 80.2 Å / 73.4 Å, 79.5 Å, 82.8 Å

In the closed Env trimer (PDB: 5ACO)[25], the distance between the Asp368 residues and Pro124 residues measured 14.7 Å and 56 Å, respectively. These distances were much larger in the CD4,17b-bound open Env trimer (PDB: 5VN3) at 76.7 Å and 84.3 Å, respectively. By contrast, in the CD4,17b,8ANC195-bound partially open Env (PDB: 6CM3), these distances were intermediate between the open and closed, at 65 Å and 79.4 Å, respectively. Since all three structures were reconstructed by imposing C3 symmetry during cryo-EM map refinement, each of these interprotomer distances was identical within each structure. These distances measured in the Population 1 structure were similar to the distances in the CD4,17b,8ANC195-bound partially open structure (PDB: 6CM3) (Fig. 3F). Since no symmetry was applied during

**Fig. 3 | Burial of FP upon gp120 opening. A** Three views of the Population 2 structure shown in cartoon representation with gp41 colored black, gp120 light gray, CD4 yellow, VRC34.01 Fab blue, 17b Fab orange. Glycans are shown in stick representation. The FP within the gp41 subunit is colored cyan. Within gp120, the bridging sheet is colored red and the **α**0 helix green. Blue circle-headed arrows indicate the gp41 positions that are not bound to VRC4.01 Fab. **B** View of Population 3 coordinates from the viral membrane shown in cartoon representation with gp41 colored black, gp120 light gray, CD4 yellow, VRC34.01 Fab blue, 17b Fab orange. Glycans are shown in stick representation. The FP within the gp41 subunit is colored cyan. **C** Population 2 structure zoomed-in view of gp41 subunit that was not bound to VRC34.01 showing the buried FP in cyan and the FPPR in light green. The EM map (EMD-46671) contoured at a level of 0.105 in ChimeraX is shown as a transparent surface with fitted coordinates shown in cartoon representation. **D** Population 3 structure zoomed-in view of its two gp41 subunits that were not bound to VRC34.01, showing the buried FP in cyan and the FPPR in light green. The EM map (EMD-46672) contoured at a level of 0.123 in ChimeraX is shown as a transparent surface with fitted coordinates shown in cartoon representation. **E**, **F** Extent of Env openness measured as the distance between residue 368 (blue spheres) and residue 124 (red spheres) in (**E**) previously published Env conformational states and (**F**). Env conformational states defined in this study.

the reconstruction of the Population 1 map, three distances were noted for each measure: 63 Å, 66 Å, and 66.9 Å for the interprotomer distances between residue Asp368, and 75 Å, 77.6 Å and 78.2 Å for the interprotomer distances between residue Pro124. In Population 2, the two protomers that were bound to VRC34.01 showed similar separation as observed in Population 1, 61.4 Å and 74.6 Å between residues Asp368 and Pro124, respectively. The protomer that was not bound to VRC34.01 showed greater gp120 displacement with these distances approaching closer to those observed in the CD4,17b-bound open Env trimer (PDB: 5VN3). In Population 3, the two protomers that were not bound to VRC34.01 had buried FPs and showed gp120 geometries closer to the fully open Env conformation.

Taken together, our results demonstrate that FP burial rendering it inaccessible to an FP-directed antibody requires further Env opening and transition of Env geometry past a symmetrically open intermediate that occurs earlier during CD4-induced Env opening. This CD4,17b,VRC34.01-bound BG505 SOSIP Population 1 (PDB: 9D90, this study) intermediate resembles the cryo-ET structure of the CD4-bound HIV-1$_{ADA.CM}$ Env[6], and the single particle cryo-EM structures of CD4,17b,8ANC195-bound BG505 SOSIP (PDB: 6CM3)[23] and CD4,17b,8ANC195-bound B41 SOSIP (PDB: 6EDU)[23]. In this early intermediate, FP can adopt a buried (antibody-inaccessible) or an antibody-accessible conformation. This intermediate is characterized by gp120 protomers opening like the petals of a tulip where the Env trimer apex separates and gp120 is displaced from the trimer central axis, as a rigid body hinging about the gp120 N/C termini at the trimer base. For stable sequestration of FP that renders it unavailable for antibody binding, further displacement of the gp120 protomers is needed, in the form of a lateral rotation in a plane roughly parallel to the viral membrane and about an axis orthogonal to the central trimer axis.

## Conformational changes in gp41 required for stable FP sequestering

To elucidate the gp41 structural features involved in stable FP sequestration, we examined differences in the vicinity of the FP between the partially open Population 1 conformation (PDB: 9D90; this study) and the previously described fully open Env (PDB 5VN3)[17]. The fully open structure showed a greater displacement of the gp120 subunits visualized by the clear separation of signature α0 helix from gp41, while in the Population 1 structure this region was helical but remained associated with the gp41 subunit (Fig. 4A).

At the FP site, the most striking difference was observed in the gp120/gp41 pocket where FP was buried in the fully open structure versus this region in the partially open intermediate (Fig. 4B, C). In the CD4,17b-bound fully open structure (PDB; 5VN3), this pocket was much larger and measured at ~26 Å between FPPR residue Gln540 and gp120 residue Phe233, with the buried FP adopting an extended loop conformation to fill the larger space of the pocket. By contrast, in the partially open intermediate, this distance measured 13 Å in Population 1 (CD4,17b, VRC34.01-bound structure) where the FP was exposed and 19 Å in the CD4,17b, 8ANC195-bound structure where the FP was buried and assumed a helical conformation. Progressive straightening of FPPR along with changes in both HR1 and HR2 regions of gp41

orchestrated the enlargement of this pocket, which in the fully open structure assumes an interprotomer character with one of its walls lined with the HR1 helix of the adjacent protomer. Thus, the concerted gp120/gp41 re-organizations that resulted in the formation of a larger FP-binding pocket may be responsible for the stable sequestration of the FP in the fully open structure.

We observed that the Population 2 gp41 with buried FP had a conformation similar to that of the fully open Env (PDB: 5VN3), with HR1 helices showing close overlap and the FPPR straightened out further compared to the partially open Population 1 and the 6CM3 structures, albeit not to the extent of the fully open structure (Fig. 4D). In both Population 2 and fully open 5VN3 structures, the movement of the HR2 region around residues 638–662 (indicated by red arrow in Fig. 4D) creates room for the FPPR unbending. The gp120/gp41 pocket in this Population 2 protomer measured 21 Å, with the cavity size approaching that of the cavity measured in the fully open structure.

In summary, our data show that the FP configuration undergoes stepwise changes as a consequence of CD4-induced movements in gp120 and gp41. From a closed, pre-receptor state where FP is accessible to antibodies, Env proceeds to partially open states where FP remains available to the FP-directed antibody VRC34.01. Only upon more extensive rotation of gp120 and concerted changes in gp41 does FP become fully buried within a gp120/gp41 pocket and, as a result, no longer accessible to antibody binding.

## smFRET analysis of Env opening on the virion surface

smFRET analysis of Env on the surface of intact virions has revealed conformational shifts of virus Env from a pre-triggered (PT) state through a pre-receptor closed (PC) state to a fully open CD4-bound conformational state (CO) in response to CD4 activation[26,27]. The pre-fusion, pre-receptor closed Env on virions resembles the FP-accessible Env structure complexed with three VRC34.01 (PDB: 5I8H), while the fully open Env was associated with the symmetric Env structure bound with three CD4 and three 17b (PDB: 5VN3), and a pre-triggered state was suggested that is undefined in currently available structures[26,27]. We asked whether the structural differences between partially open VRC34.01-bound Env structures characterized in this study and the fully open FP-sequestered Env would be reflected at the global population level of Env conformations presented on virions. We performed smFRET experiments at room temperature (~25 °C) of two different fluorescently click-labeled Env$_{BG505}$ on intact HIV-1$_{Q23}$ virions[28], in which donor/acceptor fluorescent probes were placed between gp120 V1 and V4 or between gp120 V4 and gp41 α6, respectively (Supplementary Fig. S12A). Placing FRET probes at different paired structural elements of Env allowed us to visualize global conformational changes of Env from two different structural perspectives, gp120 V1-V4 and gp120-gp41 (Fig. 4E–G, and Supplementary Fig. S12, and Table S4). Using these two imaging systems, we observed distinct FRET histograms of virus Env and similar trends of histogram shifts across different experimental conditions: ligand-free, in the presence of ligands VRC34.01, VRC34.01 + sCD4 + 17b, and sCD4 + 17b (Figs. 4E–G and S12, Table S4). The distinct FRET histograms observed from the gp120 V1-V4 (Supplementary Fig. S12B, D) and gp120-gp41 (Supplementary

**A** State of the FP along CD4-induced Env opening pathway

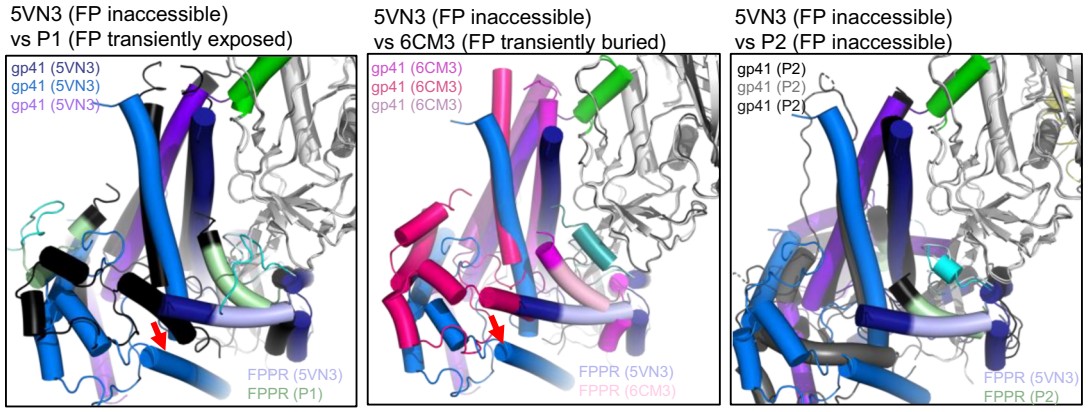

**B**

↻ 180°

**C**

| | | | | |
|---|---|---|---|---|
| BG505 SOSIP | BG505 SOSIP | BG505 SOSIP | BG505 SOSIP | B41 SOSIP |
| VRC34.01 | CD4, 17b, VRC34.01 | CD4, 17b, 8ANC195 | CD4, 17b, VRC34.01 | CD4, 17b |
| 5I8H | 9D90 | 6CM3 | EMD-46671 | 5VN3 |
| Closed | Partially open | Partially open | Partially open | Open |
| FP accessible | FP accessible | FP inaccessible | FP inaccessible | FP inaccessible |

**D** gp41 reorganization required to stably sequester FP

5VN3 (FP inaccessible) vs P1 (FP transiently exposed)

5VN3 (FP inaccessible) vs 6CM3 (FP transiently buried)

5VN3 (FP inaccessible) vs P2 (FP inaccessible)

smFRET analysis of impact of VRC34.01 binding on Env opening trajectory

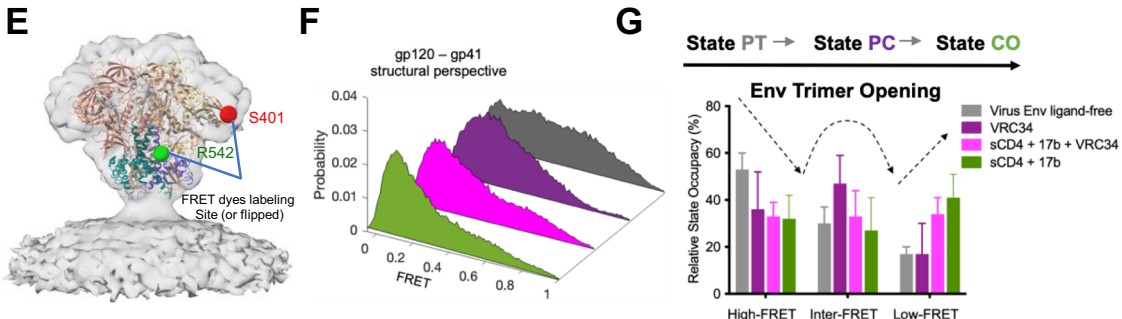

**E**

**F** gp120 – gp41 structural perspective

**G** State PT → State PC → State CO

Env Trimer Opening

■ Virus Env ligand-free
■ VRC34
■ sCD4 + 17b + VRC34
■ sCD4 + 17b

Figs. S12F and 4F) imaging systems are expected, as they capture Env dynamics from two different structural perspectives (Supplementary Figs. S12C and 4E). Due to differences in viewing angles, the FRET efficiencies associated with each primary state vary between systems. Of note, the similarity in shift directions under ligand-free and ligand-present conditions (Figs. 4E–G and S12) suggests that Env undergoes

global conformational changes at the population level, independent of the observation angle. In this analysis, we applied the previously well-defined three-state (PT, PC, CO) Gaussian distributions[28] to describe the FRET histograms, which reflect the overall conformational land-scape of Env on virions. As expected, ligand-free Env exhibited pre-dominance of the pre-triggered conformation, and Env, in the

**Fig. 4 | FP trajectory upon CD4-induced Env opening. A** HIV-1 Env structures organized by extent of opening. Left to right: closed, VRC34.01-bound Env (PDB: 5I8H) with gp41 colored olive, FPPR orange and FP cyan; partially open, CD4,17b,VRC34.01-bound Env (PDB:9D90; this study) with gp41 colored black, FPPR light green and FP cyan; partially open, CD4,17b,8ANC195-bound Env (PDB:6CM3) with gp41 colored magenta, FPPR light pink and FP teal; partially open CD4,17b,VRC34.01-bound Env (EMD-46671; this study) with gp41 colored black, FPPR light green and FP cyan; fully open, CD4,17b-bound Env (PDB:5VN3) with gp41 colored blue, FPPR light blue and FP cyan. The gp120 subunit is colored gray. Inset: Zoomed-in view of region around **α0** helix (green). **B** 180° rotated views A with brown squares around the FP (colored cyan except in partially open CD,17b,8ANC195-bound BG505 structure, where it is colored teal). **C** Zoomed-in views of region around FP. Red arrows indicate direction of CD4-induced Env opening from pre-CD4, closed Env to CD4-induced fully open Env. **D** Comparisons of gp41 organization between fully open Env with sequestered and inaccessible FP

(PDB: 5VN3), and (left) P1 (transiently exposed FP), (middle) partially open Env with transiently buried FP (PDB: 6CM3), and (right) P2 (inaccessible FP). Red arrows (left and middle panels) indicate HR2 movement that allows the FPPR to re-orient creating space for FP burial. **E–G** Locations in Env structure (PDB 4ZMJ fitted into EMDB-21412) where two fluorophores (Cy3 and Cy5 derivatives) are attached for smFRET imaging (**E**), three-dimensional presentation (**F**) and quantification (**G**) of conformational distribution-indicated FRET histograms observed from the gp120-gp41 perspective. The probability of each state (**G**), presented as mean ± s.e.m. (uncertainty), was derived from histograms (**F**), each compiled from $N_m > 200$ traces (specifically, 241, 213, 236, and 242), as detailed in Fig. S12F. The determining parameters are listed in Table S4. Virus Env$_{BG505}$ samples three primary conformational states (PT Pre-triggered, PC Prefusion Closed, and CO: CD4-bound open). PT predominates in the ligand-free condition, while VRC34 shifts the conformational landscape differently from that of the CD4-bound opening. Source data are provided as a Source Data file.Source Data.

presence of sCD4 and 17b, prevailed in the fully open CD4-bound state. In the presence of VRC34.01, a decrease was observed in the PT population with an increase in the PC population, consistent with previously published results with the JR-FL Env[7]. For the VRC34.01 + sCD4 + 17b sample, the smFRET histograms suggested that the Env conformational distributions resided between the PC and CO conformations (Fig. 4F, and Supplementary Figs. S12B, D, F). Quantifying and comparing the propensity of each primary conformational state occupied by virus Env under ligand-free and different ligand-bound conditions, we observed distinct conformational effect on Env by VRC34.01 in the presence of sCD4 + 17b, positioned on the Env activation pathway between the effect of VRC34.01 alone and the sCD4 + 17b CO state (Fig. 4G and Supplementary Fig. S12E). These results were consistent between the observations from the gp120-gp41 (Fig. 4E–G and Supplementary Fig. S12F) and the gp120 V1-V4 structural perspectives (Supplementary Figs. S12B–E). Thus, smFRET analysis of the impact of VRC34.01 on the CD4,17b-bound Env was consistent with VRC34.01 stabilizing an intermediate state on the path of CD4-induced Env opening.

**Visualizing conformations preceding CD4-induced transitions of HIV-1 Env**

We next sought to visualize CD4-induced Env conformational transitions that occur upstream to Population 1 by performing single particle cryo-EM analysis on a sample of BG505 SOSIP that was incubated with sCD4 for 2 h, followed by the addition of VRC34.01 Fab 30 min before sample vitrification. As antibody 17b works synergistically with CD4 to open Env (Fig. 1C), we rationalized that excluding 17b may allow us to capture earlier stages of CD4-induced Env conformational changes. Two distinct particle populations were revealed in the cryo-EM dataset, which yielded reconstructions of 4.08 Å (Population 4) and 4.14 Å (Population 5) global resolutions. For both populations, all three protomers were bound to one each of CD4 and VRC34.01 Fab (Fig. 5A, B and Supplementary Fig. S13, and Table S1, S3). The two populations differed in the extent of rotation of their gp120 subunits. In Population 4, one of the three gp120 protomers was rotated roughly to the extent observed in the Population 1 structure, while the other two protomers were minimally rotated from their pre-receptor conformation. The distances between the CD4 binding site residue Asp 368 in the two minimally rotated protomers measured 60.2 Å and was thus closer to the distance observed in the closed, pre-receptor Env (~56 Å) (Fig. 3E) than to the distances measured in the partially open Population 1 intermediate (~75–78.2 Å) (Fig. 3F). The third protomer that had rotated was separated in this measure from the two other protomers by 61.7 Å and 66.9 Å. The bridging sheet and the α0 helix, which are the structural components associated with CD4-induced Env opening, were only seen in the rotated gp120 protomers. Although two protomers were minimally rotated, their V1V2 regions were unstructured, thus suggesting that CD4-induced disruption of the V1V2 cap of the

closed, pre-receptor Env precedes the rotation of the protomers. This is consistent with time-resolved, temperature-jump small-angle x-ray scattering studies that suggest an order-to-disorder transition in the trimer apex precedes Env transitions involving protomer rotation[29]. In Population 5, two gp120 protomers were rotated and showed formation of the bridging sheet and α0 helix, while the third protomer remained minimally rotated. Both Populations 4 and 5 were bound to VRC34.01 Fab at all the three FP sites, as expected, since the gp120 subunits had not rotated far enough to allow the gp41 conformational changes required for FP burial and steric inaccessibility to antibodies.

In summary, Population 4 and 5 structures represent conformational states preceding the partially opened Population 1 conformation. Taken together, our results demonstrate sequential CD4-induced opening of the gp120 protomers and are consistent with previous studies that show initiation of CD4 binding to the closed Env begins with a single CD4, which induces opening of gp120 protomers needed for binding of additional CD4 molecules[16,30].

## Discussion

HIV-1 entry involves sequential receptor mediated conformational changes. Knowledge of how each part of the Env moves, in synergy with other parts, provides clues that have enabled the design of immunogens[17,31]. Despite years of intense research, critical gaps remain in our knowledge of the HIV-1 entry mechanism. In this study, we have addressed one such gap related to the fate of the FP during receptor mediated Env conformational changes. The HIV-1 FP is accessible and a target for bnAbs in the closed Env but becomes buried within a gp120/gp41 cavity in the receptor-bound fully open Env[7,17]. Here, we determine a series of structures that model a stepwise transition of the FP between these two accessible and buried steps. In Table S2 we summarize the structural differences between these Env states. We add new knowledge to a previously described functional, receptor-bound, partially open Env intermediate[6] by showing that the FP remains accessible to FP-directed antibodies (Population 1; Step 5 of Fig. 5C). Elucidating the stepwise formation of this functional intermediate (Steps 1–4, Fig. 5C), we describe two structures, Populations 4 and 5, that represent CD4-induced events earlier than Population 1, with the bridging sheet and α0 helix formed in one, two or all three gp120s, in Populations 4, 5 and 1, respectively. We describe the conformational changes downstream of Population 1, where sequential loss of FP accessibility and reduction of VRC34.01 stoichiometry occurs in one protomer in Population 2 and in two protomers in Population 3, as the gp120 subunits in these protomers undergo further lateral rotation to lead to the fully open Env[17] where all gp120s are fully rotated and FP in all protomers stably sequestered and unable to bind to VRC34.01 (Steps 6–8, Fig. 5C). From its receptor-bound, fully open conformation, further conformational changes are needed for Env to release its fusion peptide for insertion into the host membrane. Future studies

**A** Population 4 bound to CD4 and FP-directed antibody VRC34.01

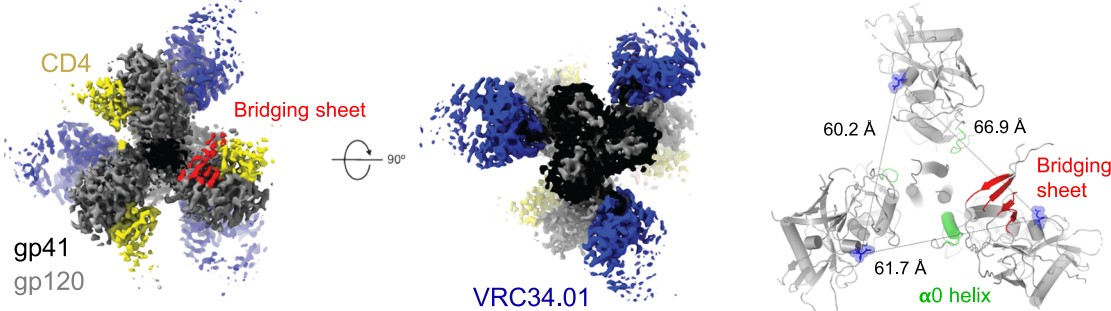

**B** Population 5 bound to CD4 and FP-directed antibody VRC34.01

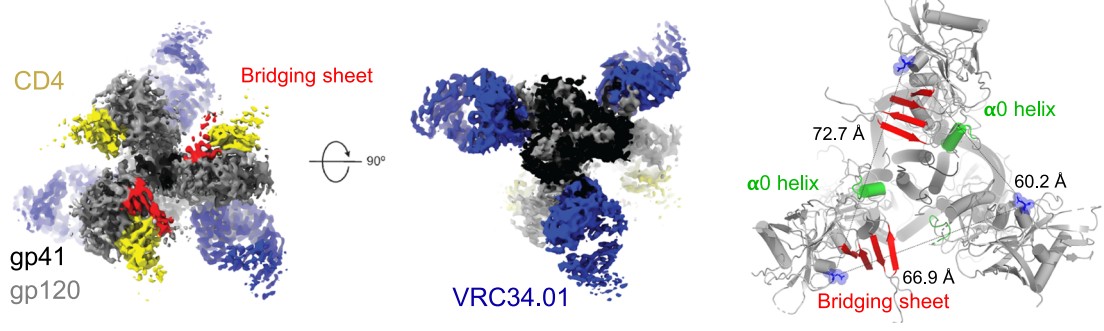

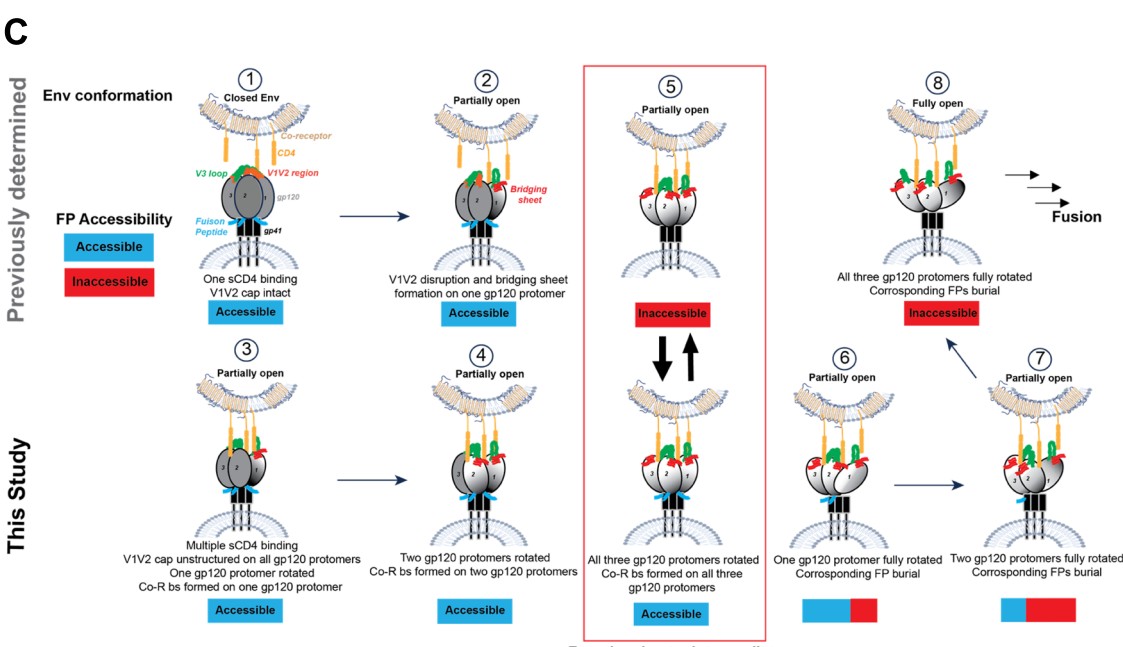

will reveal how other FP-targeting antibodies interact with the receptor-bound Env intermediates and whether they recapitulate the interactions made by VRC34.01. Additional studies will also be needed to address the generality of the observed intermediates across different HIV-1 isolates.

The FP site of vulnerability on HIV-1 is one of the few sites of Env vulnerability against which vaccination has succeeded in elicited antibodies of over 50% neutralization breadth[32]. Like VRC34.01, these antibodies recognize the accessible conformation of FP in the prefusion-closed state, with the revealed mechanistic details indicating antibody recognition of FP extends into the early entry intermediates identified in this study. Thus, in both the prefusion-closed state as well

as early entry intermediates, FP appears to have considerable conformational flexibility, which is crucial for vaccine priming with flexible peptides linked to carriers[10,33].

From being solvent exposed and accessible to antibodies to becoming transiently, then stably sequestered in an antibody inaccessible conformation, to finally being released from this sequestered state to insert into the host membrane and mediate fusion, the HIV-1 FP follows a trajectory that is unique among Type 1 fusion proteins. The typical norm for other viruses is to hold their FP in a partially or wholly occluded, sometimes metastable conformation in the pre-receptor state, which receptor binding mediated conformational changes then release for insertion into the host membrane[34]. For SARS-CoV-2, for

**Fig. 5 | Partially open early intermediates and a stepwise mechanism for CD4-induced Env opening.** **A** CD4,VRC34.01-bound BG505 SOSIP Env with rotation, bridging sheet (red) and **α**0 helix (green) formation observed in a single gp120. **B** CD4,VRC34.01-bound BG505 SOSIP Env with rotation, bridging sheet (red) and **α**0 helix (green) formation observed in two gp120 subunits. **C** A structure-guided mechanism for stepwise Env opening along the HIV-1 entry pathway. Top panel structures were determined previously, bottom panel structures were determined in this study. Stepwise transitions are marked with numbers within a circle on top of each structure starting from (1) the binding of a single CD4 to a closed Env trimer (PDB: 5U1F, 8FYI). This is followed by (2) opening of the Env trimer (EMD-29292) that allows additional CD4 molecules to bind. (5) A partially open Env conformation was described bound to CD4, a coreceptor (Co-R) mimicking antibody, and the gp120/gp41 interface targeting antibody 8ANC195 (PDB: 6CM3, 6EDU) where the FP was buried within a gp41 cavity. The CD4-induced opening of the HIV-1 Env

culminates in the complete rotation of all the gp120 subunits that are accompanied by gp41 conformational changes and resulting in the burial of FP. This state is numbered (8) in this schematic. This study showed that the geometry of the functional entry intermediate (5) that was also visualized on membrane-associated Env (EMD-29294), was compatible with the FP being either buried or exposed, and thus, in this conformation the FP was accessible to antibodies. Further, this study filled in mechanistic gaps between (2) and (5) by showing stepwise gp120 rotations to reach this functional entry intermediate. Finally, this study visualized a stepwise mechanism for how the functional entry intermediate (5) may transition to the fully open Env (8), yet again by stepwise opening of the each gp120 subunit from its partially rotated to the fully rotated conformation, which was accompanied by burial of the FP in the corresponding protomer. Schematics of membranes and the host receptors were created in BioRender. Acharya, P. (2025) https://BioRender.com/7tpbllk.

example, FP-directed antibodies have been described but these antibodies cannot access their FP epitopes in the pre-receptor conformation of the spike protein[35–38]. Receptor binding mediated conformational changes reveal these cryptic epitopes to allow antibody binding[36,39,40]. In summary, the HIV-1 FP tracks a unique trajectory among Type 1 fusion proteins with its multistep receptor-induced conformational transitions. Despite these differences, the central role of receptor-induced conformational changes in controlling and maneuvering the FP through its pre-receptor conformation to its fusion competent state is a common feature amongst all Type 1 fusion proteins.

One apparent conundrum this study elucidates is the genesis of the FP as a site of vulnerability in HIV-1. Why, in a virus that has evolved exemplary defenses to shield its vulnerabilities from the immune system, would the highly conserved FP be exposed and a focus for targeting by broadly neutralizing antibodies? Why would it not be hidden from the immune system within the pre-receptor, closed Env? In this study, we have shown that at least partial opening of the HIV-1 Env trimer is required for burial of FP within a gp41 cavity that forms because of this opening. While a partially open Env can accommodate FP burial, this study demonstrated that such burial is not stable, and only upon more substantial Env opening does FP become stably sequestered. Since opening of Env, even partial, exposes epitopes that make the virus more susceptible to neutralization, on balance, it may have been advantageous for maintaining the neutralization resistant compact and closed HIV-1 Env conformation to leave the FP exposed, and shield it with a few strategically placed glycans. The exposure of the FP, and thus the creation of the site of vulnerability to antibody, may be a mitigating step towards preventing greater vulnerability due to Env opening that may accompany burial of the flexible hydrophobic FP.

Another partially open Env conformation that has been described is the open-occluded conformation that is sampled in a receptor-independent manner by both the native Env as well as Env ectodomain constructs used for vaccine applications[41]. A recent study combining MD simulations and smFRET analysis has proposed the open-occluded conformation as neutralization-relevant intermediate of Env on the transition trajectory[42]. Structures of the open-occluded Env ectodomain show FP to be buried within a gp41 cavity[41]. Indeed, the hydrophobic solvent-exposed FP may be a source of Env metastability and its proclivity to shield itself, even if transiently, within a hydrophobic pocket may provide the underlying rationale for the accessibility of partially open Env conformations that allow such FP occlusion.

In summary, our study provides a stepwise mechanism for receptor-induced opening of HIV-1 Env and elucidates the trajectory of the fusion peptide from its solvent-exposed configuration in the pre-receptor, closed Env to its buried, antibody-inaccessible configuration in the receptor-bound, fully open Env. The use of SOSIP-stabilized Env ectodomain constructs has enabled high resolution structures of receptor bound Env intermediate states in this and previous studies[16,17,21,23,41]. While incorporation of stabilizing mutations may

impact Env opening, interpreting these high-resolution structures together with structural and spectroscopic studies on virion associated Env demonstrates the physiological relevance of key Env intermediates identified using SOSIP-stabilized Envs[6,26]. Collective evidence in our study performed using the SOSIP-stabilized Env ectodomain from the BG505 isolate suggests that the structure of Population 1 represents a general intermediate on the HIV-1 entry pathway and that this intermediate is accessible for binding to broadly neutralizing antibodies such as VRC34.01 and 8ANC195. The evidence includes the recurrence of similar geometry in structures obtained from different isolates (BG505 or B41) and in complex with different gp120/gp41 targeting antibodies (8ANC195 and VRC34.01), and their concurrence with the cryo-ET resolved structure of the CD4-bound HIV-1ADA.CM Env in the membrane context. Our work thus reveals a functional entry intermediate (Population 1) with subunit geometry compatible with both a solvent exposed, antibody accessible FP, and an occluded, antibody inaccessible FP. By elucidating the accessibility of FP during receptor-induced Env conformational changes our study reveals key insights into this critical component of the HIV-1 entry machinery, which is a major target site for vaccine development.

## Methods
### Protein expression and purification
HIV-1 Env ectodomain constructs used in this study were purified form HEK293S GnT1- cells (Thermo Fisher Scientific) diluted at the time of transfection to $1.25 \times 10^6$ cells/mL. Before transfection, cells were diluted in Freestyle™ 293 Expression Medium (Cat No. 12338018) to $1.25 \times 10^6$ cells/mL at a volume of 950 mL. Plasmid DNA expressing the Env ectodomain and furin were co-transfected at a 4:1 ratio (650 µg and 150 µg per transfection liter, respectively) and incubated with 293fectin™ transfection reagent (ThermoFisher Cat No. 12347019) in Opti-MEM I Reduced Serum Medium (ThermoFisher Cat No. 31985062). The diluted mixture was added to the cell culture which was incubated at 37 °C, 9% $CO_2$ on a shaker at 120 rpm for 6 days. On day 6 the cell supernatant was harvested by centrifuging the cell culture at 4000 x $g$ for 30 min. The supernatant was filtered with a 0.45 µm PES filter and concentrated to approximately 100 mL using a Vivaflow® 200 cross-flow cassette (Sartorius Cat No. VF20P2).

The cell culture supernatant was passed through 10 mL PGT145 IgG-conjugated affinity column equilibrated in 20 mM PBS, pH 7.5. Following loading and washing, Env trimers were eluted using 3 M $MgCl_2$, pH 7.2. The eluted Env were concentrated to -1 mL with a Centricon-70 100 kDa filter (Millipore Sigma). After concentrating, Env were filtered through 0.22 µM filter to remove any aggregates before loading on Superose 10/300 GL column (Cytiva) size exclusion column pre-equilibrated in PBS on an AKTA Pure (Cytiva) system. The fractions corresponding to the Env trimers were pooled, concentrated, flash frozen in liquid nitrogen frozen for long-term storage at −80 °C.

Antibodies were produced in Expi293 cells and purified using a Protein A affinity column followed by size exclusion chromatography

using a HiLoad Superdex 200 column equilibrated in 20 mM PBS, pH 7.5, 0.002% w/v Azide.

4-domain CD4 was produced in Expi293 cells and purified by Q425-affinity chromatography, followed by size exclusion chromatography using a HiLoad Superdex 200 column equilibrated in 20 mM PBS, pH 7.5, 0.002% w/v Azide.

## Surface Plasmon Resonance

Surface Plasmon Resonance binding assays were performed on a T-200 Biacore system (GE-Healthcare) operating at either 25 °C or 37 °C. HBS-EP+ (10 mM HEPES, pH 7.4, 150 mM NaCl, 3 mM EDTA and 0.05% surfactant P-20) was used as running buffer. A 40 nM solution of BG505.SOSIP prepared in the running buffer was incubated at 25 °C or at 37 °C with either 200 nM of CD4, or with 200 nM CD4 and 200 nM 17b Fab. All samples were pre-incubated at the indicated temperature before mixing. After mixing, the samples were rapidly transferred to the Biacore T-200 temperature-controlled sample chamber which was pre-warmed to 25 °C or 37 °C. The samples were kept within the temperature-controlled sample chamber for the duration of the experiment. The binding surface was prepared by flowing 100 nM each of, 17b IgG and VRC34.01 IgG over each flow cells 2 and 4, respectively at 10 μl/min flow rate for 30 s with the 1st and 3rd flow cells serving as reference for 2nd and 4th flow cells, respectively. After surface pre-paration, the analyte (either BG505 SOSIP alone or BG505 SOSIP with CD4 or BG505 SOSIP with CD4 and 17b Fab) was flowed at 30 μl/min flow rate for 60 s. The same injections were carried out using HBS-EP+ buffer to obtain a reference curve. The sensorgrams were blank corrected in the Biacore T-200 evaluation software.

## Cryo-EM

Purified BG505 SOSIP.664 trimer sample stocks were diluted to a concentration of 1.3 mg/mL and were incubated with five molar excess of 4D CD4 and 5-molar excess of 17b Fab. After mixing, the samples were incubated at 25 °C or 37 °C for different incubation times. VRC34.01 Fab in 5-fold molar excess concentration was added 30 min before freezing grids. To prevent interaction of the trimer complexes with the air-water interface during vitrification, the samples were incubated in 0.085 mM n-dodecyl β-D-maltoside (DDM). A 3.5-μL drop of protein was deposited on a Quantifoil-1.2/1.3 grid (Electron Microscopy Sciences, PA) that had been glow discharged for 10 s using a PELCO easiGlow Cleaning System (Ted Pella). After a 30 s incubation in >95% humidity in a chamber that was maintained at either 25 °C or 37 °C, excess protein was blotted away for 2.5 s before being plunge frozen into liquid ethane using a Leica EM GP2 plunge freezer (Leica Microsystems). Frozen grids were imaged in a Titan Krios microscope (Thermo Fisher) equipped with a K3 detector (Gatan), using the Latitude software. The cryoSPARC (Punjani et al.) software was used for data processing[43]. Raw movies were motion corrected using Patch Motion Correction and Contrast Transfer Function (CTF) were estimated. Micrographs with CTF estimates greater than 8 Å were discarded. Automated blob picker software was used to assign the particle position, and the particles were extracted with the 320-pixel extraction box size Fourier cropped to 80 pixels. Following particle extraction, multiple rounds of 2D classification was performed to remove junk particles and re-extraction of clean particles with 320-pixel box size. A reference free ab-initio 3D reconstruction was used to create 3D reconstructions representing diverse conformational states of the Env. Further, multiple rounds of heterogeneous refinement was performed to get rid of the noise. Finally, non-uniform refinement was used on the clean particle set to obtain high resolution cryo-EM map.

Phenix, Coot, Pymol, Chimera, ChimeraX and Isolde were used for model building and refinement[44–49].

## Virus packaging and fluorescent labeling

The methods of packaging and fluorescent labeling of replication-incompetent amber-free HIV-1$_{Q23}$ viral particles with incorporated Env$_{BG505}$ have been described previously[28]. HIV-1 virions that lack reverse transcriptase (ΔRT) were prepared and used for imaging. Amber-free HIV-1$_{Q23}$ virions incorporated with two different double-tagged Env were used in this study, including dual-amber N136$_{TAG}$ S401$_{TAG}$ and hybrid click/peptide V4-A1 R542$_{TAG}$. Amber-free V1V4 N136* S401* (*, unnatural amino acid - ncAA) viruses carrying click-chemistry-reactive ncAA at 136 in V1 and 401 in V4 were produced by co-transfecting HEK293T cells with a tag-free ΔRT plasmid, an Env-tagged variant N136$_{TAG}$ S401$_{TAG}$ (TAG, amber stop codon) ΔRT plasmid, and an amber suppressor plasmid tRNA$^{Pyl}$/NESPylRS$^{AF}$. The amber suppressor can express tRNA and its cognate amino acid acyl-tRNA-synthetase in HEK293T cells. ncAA TCO* (250 μM) was added to the transfection system. Similarly, V4A1 R542* viruses were prepared using the Env-tagged V4A1 (peptide A1 tag: DSLDMLEM in V4 loop) R542$_{TAG}$ ΔRT plasmid. The ratio of tag-free vs. tagged Env plasmids used during transfection was adjusted based on previously characterized Env expression levels[28] to ensure that, statistically, on average, one tagged protomer within an Env trimer on a virion was available for fluorescent labeling (enzymatically or click)[26–28,50]. 40 h post-transfection, the supernatant was harvested and filtered, then viruses were concentrated at 113,000 ×$g$ for 2 h using an ultracentrifuge. Next, the virus pellet was resuspended using the labeling buffer containing 50 mM HEPES, 10 mM MgCl$_2$, and 10 mM CaCl$_2$. The fluorescent labeling of prepared virus Env was similar to the previously described[26–28,50]. For the amber-free V1V4 N136* S401* viruses, two TCO* were fluorescently labeled by 0.1 μM tetrazine-conjugated LD555-TTZ and LD655-TTZ by click chemistry. For the amber-free V4A1 R542* viruses, the A1 peptide in V4 was labeled by LD655-CoA, 0.65 μM in the presence of enzyme AcpS (5 μM), and the TCO* in gp41 R542 were click labeled by LD555-TTZ. Dyes were customized by Lumidyne Technologies. The above reaction mixture was incubated at room temperature overnight in the dark. PEG2000-biotin was then added at a final concentration of 0.1 mg/ml to the labeled viruses, followed by 30 min incubation at room temperature. Then, the labeled viruses were further purified using a 6%–18% gradient of Opti-prep (Sigma-Aldrich) and centrifuged at 197,120 ×$g$ for 1 h at 4 °C.

## smFRET data acquisition and analysis of virus Env

All single-molecule fluorescence resonance energy transfer (smFRET) data of fluorescently labeled viruses were collected using a custom-made prism-based total internal reflection fluorescence (prism-TIRF) microscope equipped with a fluorescence signal detection system. The detailed operating manual has been described previously[28]. Briefly, the sample loading module, a streptavidin-coated PEG passivated biotin quartz imaging chamber, was cleaned with the imaging buffer, and the background fluorescence signal was removed using the high-intensity laser. The imaging buffer contains 50 mM Tris pH 7.4, 50 mM NaCl, a cocktail of triplet-state quenchers, and oxygen scavenger: 2 mM protocatechuic acid and 8 nM protocatechuic-3,4-dioxygenase. The labeled viruses were then loaded into the sample loading module. Un-immobilized viruses were removed using the imaging buffer, and the fluorescence signals were collected. Under the ligand-present experimental conditions, fluorescently labeled viruses were incubated with the indicated 0.1 mg/ml antibody/ligand (>5x above IC95) for 30 min at room temperature before imaging. All fluorescence signals were recorded simultaneously on two synchronized sCMOS cameras (Hamamatsu ORCA-Flash4.0 V3) at 25 Hz for 80 s. The smFRET data were viewed, processed, and analyzed using the SPARTAN software package[51] shared by the Scott Blanchard lab and custom MATLAB-based scripts.

Recorded 80-second movies (2000 frames/movie) were extracted as donor/acceptor fluorescence traces (time series) with

background subtracted and crosstalk corrected. The energy transfer efficiency (FRET efficiency values or simplified as FRET in figures) from the donor fluorophore to the acceptor fluorophore was calculated using $FRET = I_A/(\gamma I_D + I_A)$, in which $I_D$ and $I_A$ represent the fluorescence of the donor and acceptor, respectively, and $\gamma$ is the correlation coefficient compensating for variations in detection efficiencies. FRET traces (FRET efficiency traces) were further derived. FRET traces reflect real-time relative distance changes between donor and acceptor, resulting from the global conformational dynamics of Env. Under each experimental condition, more than 200 individual traces were included in the final FRET histogram. These included traces meet the following filter settings: 1) a single photo bleaching point (ruling out cases of multiple labeled protomers in a trimer, multiple labeled Envs on one virion, no-labeled Env on one virion; 2) sufficient signal-to-noise ratio; 3) anti-correlated feature between donor and acceptor fluorescence (indicating active Env undergoing conformational changes, ruling out inactive Env as well as Env lacking either donor or acceptor or both). We used automatic filters in combination with manual visualization to ensure that traces of molecules with only one Cy3/Cy5-labeled protomer in a trimer on a viral particle were included for further data processing. FRET traces that meet all the above-mentioned criteria were included to compile FRET histograms/distributions. FRET histograms (conformational distributions) were presented as mean ± s.e.m. and fitted into a sum of three distinct Gaussian/Normal distributions using the least-squares fitting algorithm in MATLAB. Parameters were determined based on visual inspection of all traces that exhibit state-to-state transitions and the idealization of individual traces using three-state hidden Markov modeling. Each Gaussian represented one conformational state of virus Env. The area under each Gaussian curve was further calculated and presented as mean ± s.e.m, providing an estimation of relative state occupancy, that is, the probability of the corresponding state Env occupies with associated uncertainty.

### Reporting summary

Further information on research design is available in the Nature Portfolio Reporting Summary linked to this article.

## Data availability

The cryo-EM maps and atomic models generated in this study have been deposited in the wwPDB and EMBD databases (https://www.rcsb.org, https://www.ebi.ac.uk/emdb/) under accession codes: PDB IDs, 9D90, 9D8Y, 9D98 and EMD IDs, EMD-46655 [https://www.ebi.ac.uk/pdbe/entry/emdb/EMD-46655], EMD-46671, EMD-46672, EMD-46653, EMD-46670. Previously published structures used in this study include, 5ACO, 6CM3, 5I8H, 5VN3. Source data are provided with this paper.

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

## Acknowledgements

Cryo-EM data were collected at the Duke Krios at the Duke University Shared Materials Instrumentation Facility (SMIF), a member of the North Carolina Research Triangle Nanotechnology Network (RTNN), which is supported by the National Science Foundation (award number ECCS-2025064) as part of the National Nanotechnology Coordinated Infrastructure (NNCI). This study utilized the computational resources offered by Duke Research Computing (http://rc.duke.edu; NIH 1S10OD018164-01) at Duke University. This work was supported by NIH grants R01 AI145687 (P.A.), U54 AI170752 (P.A., R.H., and M.L.), R01 AI181600 from NIH/NIAID, an R35 GM151169 from NIH/NIGMS to M.L., and the Vaccine Research Center, an Intramural Division of NIAID, NIH.

## Author contributions

P.A. conceived the project and oversaw the study. B.T. and P.A. designed binding studies and cryo-EM experiments. B.T. expressed and purified proteins, performed SPR assays, optimized specimen, prepared cryo-EM grids, collected cryo-EM data, performed map and coordinate refinement, and performed structural analysis. R.K., W.X., and M.L. performed smFRET analysis of virus Env, and K.J. assisted with cryo-EM data collection. S.S. assisted with protein purification. R.H. performed vector analysis. P.D.K. assisted with initial SPR experiments. B.T. and P.A. wrote the first draft of the manuscript. All authors reviewed and commented on the manuscript.

## Competing interests

The authors declare the following competing interests: B.T. and P.A. have applied for patents on HIV-1 Envelope modifications related to this work. The other authors declare no competing interest.
