## [Transparent Peer Review file · Nature Communications]

Conformational trajectory of the HIV-1 fusion peptide during CD4-induced envelope opening

Corresponding Author: Professor Priyamvada Acharya

Version 0:

Reviewer comments:

Reviewer #1

(Remarks to the Author)

Thakur B. et al., investigate the conformational states of the HIV-1 fusion peptide during induced spike opening. Using cryo-EM and FRET, the authors characterize different open states of the spike that are induced by CD4/17b and the accessibility of the fusion peptide to the VRC34.01 antibody. Some additional work may improve this manuscript and make it more suitable for publication.

Major comments:

A critical concern regarding the opening experiments is the incubation temperature used. The HIV-1 spike typically functions at 37C, and deriving useful information about its opening likely requires incubations at this temperature. While it is not stated in the manuscript/method section (as it should), I can only assume that the authors incubated the spike with CD4 or CD4/17b at 4C before the measurement with the SPR due to the very long and non-physiologic incubation times reported (up to 200 hours??). Will the FP remain exposed in the presence of CD4 following incubation at 37C? I think this would be a critical point to address. In this regard, these SPR experiments (Figure 1C) also show two clear phases: a fairly "fast" phase up to ~100 hours and a stationary phase thereafter. It looks like two different populations of the spike contributed to this behavior, raising uncertainty as to what exactly is happening in this experiment. The authors should also provide all sensorgrams used to generate these data as supplementary information.

The authors interpreted the binding assay shown in Figure 1D as if FP exposure, as monitored by binding of VRC34.01, is maintained when the spike fully opens, as indicated by a saturation of 17b binding. Could it be that there are two distinct spike populations in these experiments? One that opens and one that fails to open completely? In fact, the existence of two distinct populations is also suggested by the two phases shown in Figure 1C. The Authors need to find a way to address such a possibility.

The authors discuss the three different populations of the spike in their various EM analyses. However, it is not clear from the data processing figures (S1-S4) if the datasets were processed in the same way or not. Which particles were discarded during processing? Were the same volumes and number of classes applied during 3D classifications? Which 2D classes were discarded? Considering the huge reduction of the number of particles from particle-picking to final reconstructions, could it be that there are additional 'hidden' states of the spike that were eliminated during processing? The particle distribution between the different states (populations) is meaningless if the processing of the various data sets is not identical.

The authors need to demonstrate the local quality of the EM map for their analysis of the conformation of the FP (Figure 2D, S6 & S5).

The authors failed to provide a table summarizing the geometrical quality of their models (i.e., "Table 1"). Such a table is required to assess the overall quality of the structural work.

Were the FRET experiments conducted in a temperature-controlled environment? If so, what was the temperature? If this was not 37C, the question of the temperature effect on the population distribution is relevant here as well.

The control used for the FRET measurement, which is ligand-free (grey in Figure 4E), shows very broad gp120-gp41 FRET efficiencies, indicating that the spikes are sampling a range of conformations. I would assume that in the absence of CD4, the spikes will primarily be in a closed (PT) state. This data does not agree well with the distribution shown for the V1-V4.

Can the authors comment on that?

For the structural studies, the authors have used a SOSIP construct that has a few stabilizing mutations. Can these mutations somehow affect the opening of the spike? The relative orientation of gp120 and gp41? How faithfully does this construct represent the native, unmodified spike? The authors should address, at least in the discussion, the possible limitations of using such a construct.

In the supplementary figures showing the EM processing workflows, the authors need to actually portray the processing and not just provide the final results. For example, if 3D classification steps are used, the authors need to show the different volumes and the corresponding particle distributions. For the final volumes, the authors need to also provide orientational distribution plots.

The pixel size and the exposure parameters are not included in the methods.

From the PDB-validation report, the authors used a pixel size of 1.08Å. Since all of their volumes were reconstructed to resolutions far worse than 2.16Å, it is highly advised to bin (downsample) the particles when appropriate.

Minor comments:

In Figure S3, the authors do not specify the number of particles in the data set.

Table S1 is referred to in the text (line 122) but is not included. Perhaps the authors refer to part B of Figure S4?

The color coding in Figure 2D, where population 1 is compared to 6CM3, is very confusing.

The authors should indicate the level of the EM map shown in Figures 3C & 3D. Also, the authors show volumes, or EM-maps, and not “cryo-EM reconstruction,” as noted in Figure Legend 3.

In Figure 4D, in the middle panel, it looks like the secondary structure assignment for the model in red is wrong. There is a helical segment that is shaped like a horseshow. The authors may want to reconsider that.

It would be convenient if the authors could show the location of the fluorophores used for the FRET experiment on a structure of the spike.

No axis label is shown in Figure 4E.

Reviewer #2

(Remarks to the Author)

In this manuscript, “Conformational trajectory of the HIV-1 fusion peptide during CD4-induced envelope opening”, by Thakur et al., the authors describe conformational states of HIV Env that are stabilized when bound to VRC34.01, a HIV broadly neutralizing antibody specific for the fusion peptide. They show that the FP transitions from a fully VRC34 accessible state in a pre-closed Env trimer, to a FP inaccessible state by changes that occur in the gp120 subunit upon binding to host CD4 receptor. Soluble trimeric BG505 SOSIP-stabilized gp140 ectodomains were co-incubated with CD4, an antibody that stabilizes a receptor-activated beta-sheet rich surface on gp120, 17b, and VRC34. Early kinetic intermediate bound states in various stoichiometries were examined using cryoEM, structures were binned and determined. They describe a ‘newly resolved intermediate’ and show how the apex of gp120 partially rotates out and opens, causing FP to bury in a newly formed gp120-gp41 interfacial pocket.

The manuscript contains useful new and high-quality structural data that bear on the conformational transitions made by the FP and show similarities between their ‘Population 1’ structure and a previous described fusion intermediate state of virion-associated Env determined using cryoET. This reviewer thought the manuscript was ambiguous or challenging to follow in some places with multiple substructures being considered, as discussed below, so could use some clarification. However, overall, the results presented add to a growing number of structures that help to define the structural transitions HIV Env makes during receptor-engagement prior to membrane fusion and entry.

Comments

Ln 40-42 “We determined <sic> a newly resolved intermediate and <sic> reorganizations of the gp120-gp41 interface that ultimately resulted in FP burial in an antibody-inaccessible configuration.” The abstract emphasizes a new trimer conformational state. Can the authors name this state clearly and in simple terms? With a growing number of conformational states of Env it will be crucial to have an unambiguous naming system that is intuitive and easy-to-remember. Authors refer to “partially open” trimer conformations Populations 1, 2, and 3, then later, they present Populations 4 and 5. In the figures in some places they refer to whether the Fusion Peptide (FP) is accessible, but this alone does not account for the different populations. VRC34 stoichiometries may be determining the number of the Population category, rather than FP accessibility, whereas other elements, including alpha0, are also changing. It is challenging to discern what organizing principles are most significant. Subpopulations of conformational states have been described previously, others not. Can the authors please explain the thought process?

The authors describe VRC34-bound Env structures, but other FP bnAbs will presumably stabilize different conformations, which may say more about the Ab preference and less about the conformational intermediates during fusion in absence of the Ab. Can the authors please carefully address and discuss this by comparing the structure with that of other FP bnAbs to infer anything about the generalizability of their findings and about the stoichiometry and mechanism of FP burial?

Do the authors have any evidence and/or sense whether the Subpopulations, Population 1,2,3,4,5 and so on, are in an interchangeable and dynamic equilibrium of Ab bound stoichiometries involving Env trimers that are otherwise identical to

one another, or, alternatively, whether they represent fundamentally different trimer subpopulations due to pre-existing differences perhaps in their glycosylation composition that predisposes them have different stoichiometries with VRC34?

Other Comments

Ln 32 "... and assembles into a trimer" is written ambiguously. It should be stated more clearly that the heterodimers are what assemble into a trimer (of heterodimers), and not the gp120 subunits.

Ln 56-7. "FP comprises a hydrophobic stretch of about 20 amino acids..." Prior published studies on the FP have presented data indicating only 8 amino acids are solvent accessible and is arguably definitional for the FP in the vaccine field, which begs the question of how the 20 residue FP was defined. Can the authors clarify here how/why the 20 amino acid definition was used and how it differs from the 8 amino acid definition?

Ln 59. "...and thus of vaccine focus" is an odd sentence fragment. Please reword for better clarity.

Fig.2. panel C. The PDB IDs are indicated in the different plots to make comparisons between conformation with Population 1. However, without indicating what structure each PDB ID represents it is less meaningful. Can the authors please include a short table description of the structures of each of the PDB IDs included in this figure panel?

Reviewer #3

(Remarks to the Author)

The authors use structural biology approaches to address the conformational changes of HIV Env, in particular the reported transition of the fusion peptide to being solvent, and by extension, neutralizing antibody-accessible in the prefusion "closed" state, to occlusion in a newly formed pocket triggered by receptor CD4 binding and subsequent opening of the trimer "CD4-bound prefusion open state." There has been significant work done in the general field of receptor-bound structures over the years, which the authors consistently refer to and cite. Importantly, the authors of this manuscript approach it with an important question related to vaccine design: how do the conformational changes and natural "breathing" of Env trimers affect the binding and neutralization ability of bnAbs such as VRC34.01? The results provide new insight and clearly demonstrate that bnAbs like VRC34.01 can bind certain open states of Env, which was not previously appreciated. A major highlight is that the fusion peptide is accessible to neutralizing antibodies more often than has been assumed in the field, that is, not just in prefusion "closed" state but also in certain intermediate states that foreseeably occur readily in the natural dynamics of Env, even before receptor engagement. The work is presented clearly and I support the publication of this manuscript in Nature Communications after the authors have a chance to address the following points:

1) Figure 1C,D and lines 111-115: the authors comment that substantial VRC34.01 binding occurs where near saturation 17b binding has occurred. How do the authors conclude that the trimers are fully saturated with 17b? It is not immediately clear to me from that figure, and I do not see additional SPR curves in the supplemental documents. Also, just based on SPR (without the later cryo-EM structures), can one infer that all gp120s are bound by sCD4, and in the case of 17b, the stoichiometry is also near 1:1? In other words, is the residual VRC34.01 binding a result of incomplete sCD4/17b binding, or does the bnAb recognize some forms of sCD4/17b-bound Env? Yes, this is answered shortly in the subsequent section using cryo-EM but I would use more conservative phrasing when only SPR has been presented up to this point. This can simply be in the form of a transition, "SPR implied substantial VRC34.01 binding at the time-points where near saturation of 17b...to confirm this, we performed cryo-EM..."

2) Figure 1E and lines 124-128: the authors state that population 2 has two VRC34.01 Fabs per trimer. However, the map presented in Figure 1E shows that the signal for the second Fab is weaker than the first, suggesting heterogeneity in the dataset (a mixture of 1 and 2 Fab trimers). Did the authors try to classify this further? It is likely that a significant portion of the population 2 particles should be below in population 3, bringing down the number of particles in population 2. The weaker signal for the Fab constant domains in population 1 (relative to those of 17b in the same map) might also suggest a heterogeneous mix of VRC34.01 stoichiometries that can be further classified out.

3) The previous structures in the literature that the authors cite and compare to were of B41 and BG505 SOSIP (this study uses BG505 SOSIP). Do the authors expect any genotypic differences, for example if they repeated the current experiments with B41? Especially since the open state with the occluded FP in the new pocket was in the context of B41. Perhaps that is not easy to test since the B41 sequence lacks the necessary VRC34.01 epitope. But in general, for VRC34.01 sensitive viruses, are the FP transitions and intermediates expected to be similar across different genotypes?

4) Lines 198-201 and Figures C and D: this finding, even if anticipated, is very interesting and really supports the importance of this manuscript. Well done.

5) Figure 3E,F: these measurements are great and help the reader follow the text. In F, I have trouble identifying which color distance relates to which two points (the magenta and brown font in the V1-V2 base distances, and the teal and periwinkle fonts in the CD4bs distances). Perhaps the visualization triangles are accidentally omitted in the uploaded figure (the red and blue overlaid triangles that appear in 4/6 panels). something similar would be helpful here but where each line of the triangles is the color in the distance lists.

6) Lines 242-253 and figures 4C: it's also possible, related to my earlier comment, that the extended loop versus helix and

the size of the pocket could be related to genotypic differences between B41 and BG505. There is a small difference in the FP sequence at HXB2 positions 518-519 in BG505 (VF) and B41 (FI). Perhaps the authors can check and comment whether the different positions of the aromatic side chains have any impact on how the FP arranges in the pocket. Maybe not, if they also saw a similar pocket in population 2 of the BG505 data.

7) Table S1: value for "FSC threshold" should be 0.143 instead of 0.5 to match what is reported in the wwPDB validation reports.

8) Validation reports: the authors decided to model the constant domains of the Fabs, presumably because the docked in full Fab x-ray structures as the initial model. However, in the map-model overlay and other reported stats in the validation reports, it appears that there is little density or disordered density to support these regions at the recommended threshold. I recommend leaving just the Fv portions in the models as this better agrees with the maps, and should also positively impact the atom inclusion values. Specifically, PDB 9D8Y, 9D90. Same with what I believe is the second domain of sCD4 in PDB 9D98.

9) Table S1 and validation reports: minor point but the values in Table 1 are sometimes inconsistent with those in section 4 Experimental information of the validation reports (or section 1 of map only depositions). For example, EMD-46670. Table says 306,507 particles went into the final reconstruction, report lists 259,213. Table says 59.1 electron exposure, report lists 60.9. Please check the other depositions too as each reports a different dose yet Table 1 has the same value for all columns.

10) Considering that VRC34.01 can remain bound for a long time, even in open intermediate states, do the authors think that antibody engineering of VRC34.01, to increase affinity (if possible) would increase the durability? Presumably, VRC34.01 can counter the effects of CD4/17b binding up to a certain threshold.

Version 1:

Reviewer comments:

Reviewer #1

(Remarks to the Author)

The authors have addressed most of my comments. However, in the revised version of their paper, the authors present very different binding data in Figures 1c and 1d compared with the data they presented in their initial submission. In the rebuttal letter, they only state that they modified the figure legend, not the figure itself. The authors do not indicate that they performed a different experiment at room temp., raising questions regarding this data and the reason it was replaced (they do indicate a new experiment at 37C incubation, but this is included in Figure S9).

Which graphs faithfully represent the results of these experiments, the graphs that were included in the initial submission or the graphs in the revised version? If both are, why do these graphs look so different, and why didn't the authors explicitly state that they have changed the data shown? The original graph indicated a bi-phasic behavior (which I asked about, but the authors chose not to relate to it in the rebuttal letter), while the revised graph is clearly not bi-phasic. The authors need to explain this discrepancy, how and why the results have changed, and if these are indeed different experiments, why the graphs look different/the results are not reproducible enough.

Also, the EM data, in principle, relates to the original protein sample that showed a bi-phasic binding graph. If the protein batch was replaced for new experiments, is there a possibility that the different states would be populated differently?

Unfortunately, these are significant new concerns that the authors will need to address.

Minor points:

The authors need to indicate the map levels in their "zoom-in" views.

Also, they should present mesh and not surfaces (as in Figure S6).

Reviewer #2

(Remarks to the Author)

This manuscript by Thakur et al. is a resubmission. The authors describe conformational states of HIV Env stabilized when bound to VRC34.01, a HIV broadly neutralizing antibody specific for the fusion peptide. Notably, they show FP transitions from a fully VRC34 accessible state in a pre-closed Env trimer, to a FP inaccessible state by changes that occur in the gp120 subunit upon binding to the host CD4 receptor. Soluble trimeric BG505 SOSIP-stabilized gp140 ectodomains were co-incubated with CD4, an antibody that stabilizes a receptor-activated beta-sheet-rich surface on gp120, 17b, and VRC34. Early kinetic intermediate bound states in various stoichiometries were examined using cryoEM, structures were binned and determined. They show how gp120 partially rotates out and opens, upon CD4 engagement, which causes FP to bury in a newly formed gp120-gp41 interfacial pocket.

The authors were generally responsive to requests by reviewers and included clarifications requested by this reviewer about

how to keep track of and distinguish the multiple substructures being considered. Aside from a minor comment below, this reviewer has no further concerns regarding this manuscript.

Fig.5C. The text of the descriptions at the bottom of each panel in this figure are too small to read. The text should be enlarged for greater legibility.

Reviewer #3

(Remarks to the Author)

I am satisfied with the authors' responses to my comments, and for taking the time to edit the manuscript, reprocess some of the data and improve figures. I have no further comments.

Version 2:

Reviewer comments:

Reviewer #1

(Remarks to the Author)

Please present all the data for the multiple repeats of the SPR experiments. This could be as additional curves in Figure 1, or as supplementary figures. Also, please state in the figure legend how many times this experiment was repeated.

Otherwise, the authors have addressed all of my comments.

Response to Reviewers' Comments:

Reviewer #1:

Thakur B. et al., investigate the conformational states of the HIV-1 fusion peptide during induced spike opening. Using cryo-EM and FRET, the authors characterize different open states of the spike that are induced by CD4/17b and the accessibility of the fusion peptide to the VRC34.01 antibody. Some additional work may improve this manuscript and make it more suitable for publication.

Response:

We thank the reviewer for the succinct summary, as well as their detailed suggestions to improve the study.

Major comments:

A critical concern regarding the opening experiments is the incubation temperature used. The HIV-1 spike typically functions at 37°C, and deriving useful information about its opening likely requires incubations at this temperature. While it is not stated in the manuscript/method section (as it should), I can only assume that the authors incubated the spike with CD4 or CD4/17b at 4°C before the measurement with the SPR due to the very long and non-physiologic incubation times reported (up to 200 hours??). Will the FP remain exposed in the presence of CD4 following incubation at 37°C? I think this would be a critical point to address. In this regard, these SPR experiments (Figure 1C) also show two clear phases: a fairly “fast” phase up to ~100 hours and a stationary phase thereafter. It looks like two different populations of the spike contributed to this behavior, raising uncertainty as to what exactly is happening in this experiment. The authors should also provide all sensorgrams used to generate these data as supplementary information.

Response:

As described in the methods, all incubations and SPR measurements were performed at 25 °C: *“A 40 nM solution of BG505.SOSIP prepared in the running buffer was incubated at 25 °C with either 200 nM of CD4, or with 200 nM CD4 and 200 nM 17b Fab.”*

We have now also added this information to the Figure 1 legend: *“C. Surface plasmon based binding (SPR) analysis monitoring FP burial. Env was incubated at 25 °C with either sCD4 alone or with CD4 and the coreceptor mimicking antibody 17b. At different time-points after incubation, binding was measured to the fusion peptide targeting antibody VRC34.01.”*

As suggested by the reviewer, we have now performed the SPR analysis by incubating Env with CD4 37 °C for different time periods and measuring binding at 37 °C to VRC34.01 IgG or 17b IgG captured on an anti-human Fc surface (by SPR). These results are included in Figure S9 in the revised manuscript.

We have also performed cryo-EM analysis with the samples incubated at 37 °C (Figures S8 and S9). From this dataset, we obtained states that resembled the Population 1, Population 2 and Population 3 states obtained from the 37 °C datasets. We have added the results to the revised manuscript (lines 202-209 with track changes, 125-132 without track changes): *“While the experiments described above were performed by incubating Env with the ligands at 25 °C, similar trends were observed in SPR binding assays when the incubations were performed at 37*

°C (Figure S9). A cryo-EM dataset obtained by incubating BG505 SOSIP with CD4 and 17b Fab at 37 °C for 2 hr, followed by a 30 minute incubation with VRC34.01 Fab at 37 °C prior to plunge freezing yielded structures representing Populations 1, 2 and 3. The appearance of the Population 3 structure earlier than could be detected in the 25 °C cryo-EM datasets suggested that the CD4-induced Env conformational changes are more rapid at 37 °C than at 25 °C. The recurrence of these structures across independent experiments highlights the reproducibility of these structural states.”

The SPR sensorgrams used to generate the SPR time course graphs are provided in Figure S1.

The authors interpreted the binding assay shown in Figure 1D as if FP exposure, as monitored by binding of VRC34.01, is maintained when the spike fully opens, as indicated by a saturation of 17b binding. Could it be that there are two distinct spike populations in these experiments? One that opens and one that fails to open completely? In fact, the existence of two distinct populations is also suggested by the two phases shown in Figure 1C. The Authors need to find a way to address such a possibility.

Response:

We agree with the reviewer that it is not possible to definitively distinguish from the SPR measurements alone whether VRC34.01 can bind to a partially open, 17b-bound Env, or whether the measurements are indicative of populations that open and bind 17b, and others that fail to open at all and can bind VRC34.01. Instead, the ability of VRC34.01 to bind partially open, CD4/17b-bound Env is visualized and confirmed by the cryo-EM analyses that follow the SPR measurements.

We have made the following changes in the revised manuscript to address these concerns:

1. We have removed this sentence that previously appeared at the conclusion of the SPR results: *Although these overall trends of increase in 17b binding and decrease in VRC34.01 binding upon CD4 induction were as expected, we noted the retention of substantial VRC34.01 binding at the time-points where near-saturation 17b binding had occurred, suggesting that the FP remained exposed and available for antibody binding despite substantial Env opening.*
2. We have combined the section that previously reported the SPR data with the section that immediately followed it and reported the cryo-EM data. The new section is titled “FP remains accessible for antibody binding after CD4-induced Env opening”, where the cryo-EM analysis provides direct visualization that the FP remains accessible to a FP-directed antibody (VRC34.01) post CD4-binding and despite substantial Env opening.

We do not see any evidence of VRC34.01-bound closed Env trimer populations in our cryo-EM datasets. Therefore, if such populations are present, they must be very minor, and would therefore not detract from the primary finding that the FP can remain accessible for antibody binding despite substantial CD4/17b-mediated Env opening.

[Also see response to Reviewer #3]

The authors discuss the three different populations of the spike in their various EM analyses. However, it is not clear from the data processing figures (S1-S4) if the datasets were processed in the same way or not. Which particles were discarded during processing? Were the same

volumes and number of classes applied during 3D classifications? Which 2D classes were discarded? Considering the huge reduction of the number of particles from particle-picking to final reconstructions, could it be that there are additional 'hidden' states of the spike that were eliminated during processing? The particle distribution between the different states (populations) is meaningless if the processing of the various data sets is not identical.

Response:

We have now provided detailed workflows for the cryo-EM datasets (Figures S2, S4, S5, S8, S13).

In these workflows we have shown representative 2D classes, as is typically done for such figures. In response to the reviewer's question: "Which 2D classes were discarded?" we provide in the figure below (Figure R1) a representative example from one dataset (the 25 °C, 1.3 hr dataset) showing which particles are selected and which ones are discarded at the 2D classification step:

Figure R1. Selection of 2D classes: Left, all 2D classes generated from the 25 °C, 1.3 hr dataset (Figure S2). Right, selected (top) and discarded (bottom) 2D classes.

The authors need to demonstrate the local quality of the EM map for their analysis of the conformation of the FP (Figure 2D, S6 & S5).

Zoomed-in views of these regions demonstrating the local quality of the EM maps are now included in Figures S3 and S6.

The authors failed to provide a table summarizing the geometrical quality of their models (i.e., "Table 1"). Such a table is required to assess the overall quality of the structural work.

Response:

Table S1 (provided as a Microsoft Excel table) in the submitted manuscript contained details of cryo-EM data collection parameters and coordinate refinement statistics. In the revised manuscript, we have updated Table S1 and have provided this within the combined Supplemental materials ensuring that it is not inadvertently missed.

Were the FRET experiments conducted in a temperature-controlled environment? If so, what was the temperature? If this was not 37°C, the question of the temperature effect on the population distribution is relevant here as well.

Response:

The smFRET experiments were conducted at room temperature (~25°C) as our customized prism-based TIRF microscopy setup lacks temperature control. Env trimer opening/activation at room temperature has been observed in previous smFRET studies and cryoEM/cryoET across different labs (PMID:25298114, PMID: 29561264, PMID: 38232732, PMID: 37993716, PMID: 37993719, etc.), as well as in other surface fusion glycoproteins of enveloped viruses. Various factors, including temperature, time, and ligand presence, can influence conformational distributions of Env (and other biological molecules). In the presence of ligands, the preferentially recognized conformation can be enriched and stabilized. In this study, our smFRET observations of Env conformational ensembles and shifts at room temperature are well-supported by independent approaches, as they align with consistent findings from SPR and Cryo-EM studies conducted at both 37°C and room temperature.

The control used for the FRET measurement, which is ligand-free (grey in Figure 4E), shows very broad gp120-gp41 FRET efficiencies, indicating that the spikes are sampling a range of conformations. I would assume that in the absence of CD4, the spikes will primarily be in a closed (PT) state. This data does not agree well with the distribution shown for the V1-V4. Can the authors comment on that?

Response:

We thank Reviewer 1 for the opportunity to clarify. As Reviewer 1 correctly interpreted, the broad FRET distribution observed from the gp120-gp41 structural perspective indeed indicates that Env samples multiple conformations. Reviewer 1 is also right about the predominant PT state in the absence of CD4. Our data obtained from the gp120-gp41 perspective are in global agreement with those obtained from the gp120 V1V4 perspective. We have now added and highlighted clarifications in the revised manuscript. Placing fluorescent probes at different paired structural elements of Env allowed us to visualize global conformational changes of Env from two different structural perspectives, gp120 V1-V4 and gp120-gp41. Using these two imaging systems, we observed distinct FRET histograms of virus Env, and similar trends of histogram shifts across different experimental conditions: ligand-free, in the presence of ligands VRC34.01, VRC34.01+sCD4+17b, and sCD4+17b. The distinct FRET histograms from the gp120 V1-V4 and gp120-gp41 imaging systems are expected, as they capture Env dynamics from different structural perspectives. Due to differences in viewing angles, the FRET efficiencies associated with each primary state vary between systems. Of note, the similarity in shift directions under ligand-free and ligand-present conditions suggests that Env undergoes global conformational changes at the population level, independent of the observation angle.

For the structural studies, the authors have used a SOSIP construct that has a few stabilizing mutations. Can these mutations somehow affect the opening of the spike? The relative orientation of gp120 and gp41? How faithfully does this construct represent the native, unmodified spike? The authors should address, at least in the discussion, the possible limitations of using such a construct.

Response:

We have added to the discussion a section (highlighted in red font below) discussing the potential limitations of SOSIP constructs and how to ensure that the identified states are functionally relevant (ie., by comparing the results with studies done with virion associated Env):

“In summary, our study provides a stepwise mechanism for receptor-induced opening of the HIV-1 Env and elucidates the trajectory of the fusion peptide from its solvent exposed configuration in the pre-receptor, closed Env to its buried, antibody-inaccessible configuration in the receptor-bound, fully open Env. The use of SOSIP stabilized Env ectodomain constructs have enabled high resolution structures of receptor bound Env intermediate states in this and previous studies (PMID: 28700571, 37993719, 30308160, 31792452, 35136084). While incorporation of stabilizing mutations may impact Env opening, interpreting these high-resolution structures together with structural and spectroscopic studies on virion associated Env, have demonstrated the physiological relevance of key Env intermediates identified using SOSIP stabilized Envs (PMID: 37993716, 30971821). Collective evidence in our study performed using the SOSIP-stabilized Env ectodomain from the BG505 isolate suggests that the structure of Population 1 represents a general intermediate on the HIV-1 entry pathway and that this intermediate is accessible for binding to broadly neutralizing antibodies such as VRC34.01 and 8ANC195. The evidence includes the recurrence of similar geometry in structures obtained from different isolates (BG505 or B41) and in complex with different gp120/gp41 targeting antibodies (8ANC195 and VRC34.01), and their concurrence with the cryo-ET resolved structure of the CD4-bound HIV-1ADA.CM Env in the membrane context. Our work thus reveals a functional entry intermediate (Population 1) with subunit geometry compatible with both a solvent exposed, antibody accessible FP, and an occluded, antibody inaccessible FP. By elucidating the accessibility of FP during receptor-induced Env conformational changes our study reveals key insights into this critical component of the HIV-1 entry machinery and a major target site for vaccine development.”

In the supplementary figures showing the EM processing workflows, the authors need to actually portray the processing and not just provide the final results. For example, if 3D classification steps are used, the authors need to show the different volumes and the corresponding particle distributions. For the final volumes, the authors need to also provide orientational distribution plots.

The pixel size and the exposure parameters are not included in the methods.

Response:

We have provided detailed workflows for the cryo-EM datasets (Figures S2, S4, S5, S8, S13). Orientational distribution plots have been provided for each of the final deposited volumes. Pixel size and exposure parameters are listed in Table S1.

From the PDB-validation report, the authors used a pixel size of 1.08Å. Since all of their volumes were reconstructed to resolutions far worse than 2.16Å, it is highly advised to bin (downsample) the particles when appropriate.

Response:

The particles were binned 4x during the initial (junk removal) stage of data processing. This is now clarified in the cryo-EM data processing workflows.

Minor comments:

In Figure S3, the authors do not specify the number of particles in the data set.

Response:

Particle numbers have been added to the workflow for this dataset (now Figure S5).

Table S1 is referred to in the text (line 122) but is not included. Perhaps the authors refer to part B of Figure S4?

Response:

Table S1 (provided as a Microsoft Excel table) in the submitted manuscript contained details of cryo-EM data collection parameters and coordinate refinement statistics. In the revised manuscript, we have updated Table S1 and have provided this within the combined Supplemental materials ensuring that it is not inadvertently missed.

The color coding in Figure 2D, where population 1 is compared to 6CM3, is very confusing.

Response:

We have simplified the color scheme in Figure 2D. In the revised color scheme, the gp120 subunits of both the Population 1 structure and the 6CM3 structure are colored light gray, gp41 of both structures colored black with FP colored cyan in Population 1 and dark teal in 6CM3 and FPPR colored light green in Population 1 and light pink in 6CM3. The figure legend has been updated accordingly.

The authors should indicate the level of the EM map shown in Figures 3C & 3D. Also, the authors show volumes, or EM-maps, and not “cryo-EM reconstruction,” as noted in Figure Legend 3.

Response:

We have added the contour levels of each map in the Figure 3C and 3D legends. We have replaced “cryo-EM reconstruction” in the legend with “EM map”.

In Figure 4D, in the middle panel, it looks like the secondary structure assignment for the model in red is wrong. There is a helical segment that is shaped like a horseshow. The authors may want to reconsider that.

Response:

We have fixed this in Figure 4D, middle panel.

It would be convenient if the authors could show the location of the fluorophores used for the FRET experiment on a structure of the spike.

Response:

We appreciate this suggestion and have now added figures (Figure 4E for the gp120 V1-V4 angle and Figure S12C for the gp120-gp41 angle) to indicate the fluorophore attachment sites on the Env trimer structure.

No axis label is shown in Figure 4E.

Response:

Thank you for pointing out the missing axis label. We have added the axial label to Figure 4F (previously Figure 4E) in the revised manuscript.

Reviewer #2:

In this manuscript, “Conformational trajectory of the HIV-1 fusion peptide during CD4-induced envelope opening”, by Thakur et al., the authors describe conformational states of HIV Env that are stabilized when bound to VRC34.01, a HIV broadly neutralizing antibody specific for the fusion peptide. They show that the FP transitions from a fully VRC34 accessible state in a pre-closed Env trimer, to a FP inaccessible state by changes that occur in the gp120 subunit upon binding to host CD4 receptor. Soluble trimeric BG505 SOSIP-stabilized gp140 ectodomains were co-incubated with CD4, an antibody that stabilizes a receptor-activated beta-sheet rich surface on gp120, 17b, and VRC34. Early kinetic intermediate bound states in various stoichiometries were examined using cryoEM, structures were binned and determined. They describe a ‘newly resolved intermediate’ and show how the apex of gp120 partially rotates out and opens, causing FP to bury in a newly formed gp120-gp41 interfacial pocket.

Response:

We thank the reviewer for the succinct summary of the main findings of this study.

The manuscript contains useful new and high-quality structural data that bear on the conformational transitions made by the FP and show similarities between their ‘Population 1’ structure and a previous described fusion intermediate state of virion-associated Env determined using cryoET. This reviewer thought the manuscript was ambiguous or challenging to follow in some places with multiple substructures being considered, as discussed below, so could use some clarification. However, overall, the results presented add to a growing number of structures that help to define the structural transitions HIV Env makes during receptor-engagement prior to membrane fusion and entry.

Response:

We appreciate the reviewer’s favorable comments about the novelty, quality and the usefulness of the structures described in this manuscript and their candid assessment that “the manuscript was ambiguous or challenging to follow in some places”. We thank them for their detailed suggestions for improving the clarity of the manuscript and have addressed these as described below.

Comments

Ln 40-42 “We determined <sic> a newly resolved intermediate and <sic> reorganizations of the gp120-gp41 interface that ultimately resulted in FP burial in an antibody-inaccessible configuration.” The abstract emphasizes a new trimer conformational state. Can the authors name this state clearly and in simple terms? With a growing number of conformational states of Env it will be crucial to have an unambiguous naming system that is intuitive and easy-to-remember.

Response:

We thank the reviewer for pointing out the lack of clarity in the abstract. We have revised the abstract (see paragraph below). Instead of referring to an ambiguous “newly resolved intermediate”, we now clearly state the primary finding of this study: “*Here, using SOSIP-stabilized Env ectodomains⁵, we visualized that the FP remains accessible for antibody binding despite substantial receptor-induced Env opening.*”. Next, by clarifying that this state resembles a previously described intermediate state observed in virion-associated Env (PMID: 37993716), we highlighted the functional relevance of our finding that the FP is antibody accessible in this Env intermediate:

Modified abstract:

The hydrophobic fusion peptide (FP), a critical component of the HIV-1 entry machinery, is located at the N terminus of the envelope (Env) gp41 subunit¹⁻³. The receptor-binding gp120 subunit of Env forms a heterodimer with gp41. The gp120/gp41 heterodimer assembles into a homotrimer, in which FP is accessible for antibody binding³. Env conformational changes or “opening” that follow receptor binding result in FP relocating to a newly formed interprotomer pocket at the gp41-gp120 interface where it is sterically inaccessible to antibodies⁴. The mechanistic steps connecting the entry-related transition of antibody accessible-to-inaccessible FP configurations remain unresolved. Here, using SOSIP-stabilized Env ectodomains⁵, we visualized that the FP remains accessible for antibody binding despite substantial receptor-induced Env opening. We delineated stepwise Env opening from its closed state to a functional CD4-bound symmetrically open Env⁶ in which we showed that FP was accessible for antibody binding. We defined downstream re-organizations that resulted in the formation of a gp120/gp41 cavity into which the FP buries to become inaccessible for antibody binding. These findings improve our understanding of HIV-1 entry and delineate the entry-related conformational trajectory of a key site of HIV vulnerability to neutralizing antibody.

Authors refer to “partially open” trimer conformations Populations 1, 2, and 3, then later, they present Populations 4 and 5. In the figures in some places they refer to whether the Fusion Peptide (FP) is accessible, but this alone does not account for the different populations. VRC34 stoichiometries may be determining the number of the Population category, rather than FP accessibility, whereas other elements, including alpha0, are also changing. It is challenging to discern what organizing principles are most significant. Subpopulations of conformational states have been described previously, others not. Can the authors please explain the thought process?

Response:

We thank the reviewer for pointing out the lack of clarity in the description of the structural states described in this manuscript. As stated in lines 116-114 (142-144 with track changes on), Populations 1, 2 and 3 were differentiated at the first level by their stoichiometries of bound VRC34.01 Fab: “*We identified three particle populations across the three cryo-EM datasets that differed in their stoichiometries of bound VRC34.01 Fab (Figure 1E and S7, Table S2).*” The gp120 structural markers of CD4-induced transition, namely the bridging sheet and $\alpha 0$, were

present in all three protomers in Populations 1, 2 and 3, hence were not distinguishing features between them. The underlying cause of the decreased VRC34.01 stoichiometries in Populations 2 and 3 was sequestration of the FP within a gp120/gp41 pocket formed because of lateral rotation of the gp120 subunits about a plane roughly parallel to the viral membrane in the protomer(s) that failed to bind VRC34.01 (Figures 3C and D). Thus, Populations 1, 2 and 3, differ by, (1) bound VRC34.01 stoichiometries, (2) extent of FP sequestration, and (3) extent of gp120 rotation, with (1) being the primary defining phenotype for the purpose of this study, and (2) and (3) were what caused it.

Populations 4 and 5 are so named since they follow Populations 1, 2 and 3 in the manuscript. They represent earlier conformational changes (between the closed state and Population 1), thus they are bound to VRC34.01 at all three of their sites. What distinguishes them from each other and from Population 1 are the structural markers of CD4-induced conformational changes in the gp120 subunits, namely the bridging sheet, the $\alpha 0$ helix, and the first rotation (described in lines 229-231 (lines 318-320 with track changes on) as “opening like the petals of a tulip where the Env trimer apex separates and gp120 is displaced from the trimer central axis”). In Population 4, one of three gp120s show these structural changes, in Population 5, two out of three, and in Population 1, all three gp120s.

We realize that there is a “lot going on” and some additional clarification would be useful. We have now added a Table S2 that lists the structural changes associated with each Population, as well as their place or sequence on the CD4-induced structural change pathway:

Table S2. Summary of structural states identified in this study.

	Population 4	Population 5	Population 1	Population 2	Population 3
	CD4, VRC34.01	CD4, VRC34.01	CD4, 17b, VRC34.01	CD4, 17b, VRC34.01	CD4, 17b, VRC34.01
Ligands					
VRC34.01 bound	✓✓✓	✓✓✓	✓✓✓	✓✓X	✓XX
Bridging sheet formed	✓XX	✓✓X	✓✓✓	✓✓✓	✓✓✓
$\alpha 0$ helix formed	✓XX	✓✓X	✓✓✓	✓✓✓	✓✓✓
Fusion peptide accessible for antibody binding	✓✓✓	✓✓✓	✓✓✓	✓✓X	✓XX
Extent of receptor-induced Env conformational change					

✓/X indicates structural state per protomer

Additionally, to bring it all together, we have added the following description in the discussion in lines 346-356 (lines 472-482 with track changes on): “*In Table S2 we summarize the structural differences between these Env states. We add new knowledge to a previously described functional, receptor-bound, partially open Env intermediate by showing that the FP remains accessible to FP-directed antibodies (Population 1; Step 5 of **Figure 5C**). Elucidating the stepwise formation of this functional intermediate (Steps 1-4, **Figure 5C**), we describe two structures, Populations 4 and 5, that represent CD4-induced events earlier than Population 1, with the bridging sheet and $\alpha 0$ helix formed in one, two or all three gp120s, in Populations 4, 5 and 1, respectively. We describe the conformational changes downstream of Population 1, where sequential loss of FP accessibility and reduction of VRC34.01 stoichiometry occurs in one protomer in Population 2 and in two protomers in Population 3, as the gp120 subunits in these protomers undergo further lateral rotation to lead to the fully open Env where all gp120s are fully rotated and FP in all protomers stably sequestered and unable to bind to VRC34.01 (Steps 6-8, **Figure 5C**).*”

The authors describe VRC34-bound Env structures, but other FP bnAbs will presumably stabilize different conformations, which may say more about the Ab preference and less about the conformational intermediates during fusion in absence of the Ab. Can the authors please carefully address and discuss this by comparing the structure with that of other FP bnAbs to infer anything about the generalizability of their findings and about the stoichiometry and mechanism of FP burial?

Response:

Future studies will reveal how different FP bnAbs, depending on their binding affinities and orientations, interact with the HIV-1 Env intermediates. Whether and how they recapitulate the intermediates described in this study remains to be seen. We have now added a sentence to the discussion as follows: *“Future studies will reveal how other FP-targeting antibodies interact with the receptor-bound Env intermediates and whether they recapitulate the interactions made by VRC34.01 as described in this study.”*

The generality of the Population 1 structure and its accessibility to different bnAbs is, however, demonstrated by its interactions with VRC34.01 (this study), 8ANC195 (PMID: 30308160) and its occurrence in CD4-bound virion-associated Env (PMID: 30971821).

Do the authors have any evidence and/or sense whether the Subpopulations, Population 1,2,3,4,5 and so on, are in an interchangeable and dynamic equilibrium of Ab bound stoichiometries involving Env trimers that are otherwise identical to one another, or, alternatively, whether they represent fundamentally different trimer subpopulations due to pre-existing differences perhaps in their glycosylation composition that predisposes them have different stoichiometries with VRC34?

Response:

The differences in VRC34.01 stoichiometry between the different populations/structures described in this study arise because of CD4-induced conformational changes and are *not* caused by fundamentally different trimer subpopulations due to pre-existing differences such as in their glycosylation composition that may predisposes them have different stoichiometries with VRC34.01. The evidence for this are:

- (1) Using the same BG505 SOSIP protein, the datasets that yielded Populations 4 and 5 did not show any Env populations with sub stoichiometric VRC34.01 bound (see Figure S13). All Env classes identified in the dataset were bound to VRC34.01 Fab at all three sites, indicating that there were no differences in Env trimer subpopulations that would cause different VRC34.01 binding stoichiometries.
- (2) VRC34.01 failed to bind to those, and only those, Env protomers where the gp120 subunit was further rotated to create a gp120/gp41 interfacial pocket that buried the hydrophobic fusion peptide, rendering it inaccessible for antibody binding (see Figures 3C and D). Taken together, these evidence support a conformational mechanism for the observed reduction in VRC34.01 stoichiometry.

Other Comments

Ln 32 "... and assembles into a trimer" is written ambiguously. It should be stated more clearly that the heterodimers are what assemble into a trimer (of heterodimers), and not the gp120 subunits.

Response:

We have now rephrased as follows:

"The receptor-binding gp120 subunit of Env forms a heterodimer with gp41. The gp120/gp41 heterodimer assembles into a homotrimer, in which FP is accessible for antibody binding.

Ln 56-7. "FP comprises a hydrophobic stretch of about 20 amino acids..." Prior published studies on the FP have presented data indicating only 8 amino acids are solvent accessible and is arguably definitional for the FP in the vaccine field, which begs the question of how the 20 residue FP was defined. Can the authors clarify here how/why the 20 amino acid definition was used and how it differs from the 8 amino acid definition?

Response:

While the reviewer is correct that the N terminal end of the FP is the part that is accessible to antibodies and is the vaccine-relevant portion, the FP, by definition, is the region that inserts into the host membrane. It is the hydrophobic N terminal stretch followed by the Fusion Peptide Proximal Region (FPPR). For greater clarity and accuracy, we have revised the sentence as follows (and have provided references for FP definition):

"At the trimer base, FP comprises a hydrophobic stretch of 15-20 amino acids at the gp41 N terminus (PMID: 27174988, 39670799, 33871352). FP is a site of vulnerability to broadly neutralizing antibodies (bnAbs) and thus is a focus of vaccine development efforts (PMID: 27174988, 29867235, 31348886)."

Ln 59. "...and thus of vaccine focus" is an odd sentence fragment. Please reword for better clarity.

Response:

We have now rephrased as follows:

"FP is a site of vulnerability to broadly neutralizing antibodies (bnAbs) and thus is a focus of vaccine development efforts."

Fig.2. panel C. The PDB IDs are indicated in the different plots to make comparisons between conformation with Population 1. However, without indicating what structure each PDB ID represents it is less meaningful. Can the authors please include a short table description of the

structures of each of the PDB IDs included in this figure panel?

Response:

We have now added a Table S3 that lists all the structures used for the vector analysis in Figure 2 panel C.

Reviewer #3 (Remarks to the Author):

The authors use structural biology approaches to address the conformational changes of HIV Env, in particular the reported transition of the fusion peptide to being solvent, and by extension, neutralizing antibody-accessible in the prefusion "closed" state, to occlusion in a newly formed pocket triggered by receptor CD4 binding and subsequent opening of the trimer "CD4-bound prefusion open state." There has been significant work done in the general field of receptor-bound structures over the years, which the authors consistently refer to and cite. Importantly, the authors of this manuscript approach it with an important question related to vaccine design: how do the conformational changes and natural "breathing" of Env trimers affect the binding and neutralization ability of bnAbs such as VRC34.01? The results provide new insight and clearly demonstrate that bnAbs like VRC34.01 can bind certain open states of Env, which was not previously appreciated. A major highlight is that the fusion peptide is accessible to neutralizing antibodies more often than has been assumed in the field, that is, not just in prefusion "closed" state but also in certain intermediate states that foreseeably occur readily in the natural dynamics of Env, even before receptor engagement. The work is presented clearly and I support the publication of this manuscript in Nature Communications after the authors have a chance to address the following points:

Response:

We thank the reviewer for summarizing the main findings and the conceptual advances made by this study.

1) Figure 1C,D and lines 111-115: the authors comment that substantial VRC34.01 binding occurs where near saturation 17b binding has occurred. How do the authors conclude that the trimers are fully saturated with 17b? It is not immediately clear to me from that figure, and I do not see additional SPR curves in the supplemental documents. Also, just based on SPR (without the later cryo-EM structures), can one infer that all gp120s are bound by sCD4, and in the case of 17b, the stoichiometry is also near 1:1? In other words, is the residual VRC34.01 binding a result of incomplete sCD4/17b binding, or does the bnAb recognize some forms of sCD4/17b-bound Env? Yes, this is answered shortly in the subsequent section using cryo-EM but I would use more conservative phrasing when only SPR has been presented up to this point.

Response:

We agree with the reviewer that it is not possible to infer from the SPR data alone whether VRC34.01 is binding to sCD4/17b-bound Env or if the residual VRC34.01 binding is a result of incomplete sCD4/17b binding. As the reviewer correctly points out, this is answered in the cryo-EM section that follows. In the revised manuscript, we have removed this sentence that previously appeared at the conclusion of the SPR results: *Although these overall trends of increase in 17b binding and decrease in VRC34.01 binding upon CD4 induction were as expected, we noted the retention of substantial VRC34.01 binding at the time-points where*

near-saturation 17b binding had occurred, suggesting that the FP remained exposed and available for antibody binding despite substantial Env opening.

Additionally, we have combined the SPR results section with the cryo-EM section. Thus, the conclusion that VRC34.01 can bind partially open sCD4,17b-bound Env now follows the cryo-EM results.

[Also see response to Reviewer #1]

2) Figure 1E and lines 124-128: the authors state that population 2 has two VRC34.01 Fabs per trimer. However, the map presented in Figure 1E shows that the signal for the second Fab is weaker than the first, suggesting heterogeneity in the dataset (a mixture of 1 and 2 Fab trimers). Did the authors try to classify this further? It is likely that a significant portion of the population 2 particles should be below in population 3, bringing down the number of particles in population 2.

Response:

Spurred by this reviewer suggestion, we revisited the data processing workflow for this dataset. See Figure S5 for details of the workflow of the dataset from which the map shown for Population 2 in Figure 1E was obtained. We were able to obtain a Population 2 class where the two VRC34.01 Fabs were more equally defined. We have now replaced the Population 2 map in Figure 1E with this new map (Figure R2). We have replaced the deposited map for Population 2 with the new map.

Figure R2. Population 4 EM map.

The weaker signal for the Fab constant domains in population 1 (relative to those of 17b in the same map) might also suggest a heterogenous mix of VRC34.01 stoichiometries that can be further classified out.

Response:

Multiple iterations of further classifying Population 1 did not result in separation of any populations with compositional variation. This suggests that the relative disorder observed in the VRC34.01 Fab constant domains in Population 1 are resulting from conformational heterogeneity and differential domain flexibility.

3) The previous structures in the literature that the authors cite and compare to were of B41 and BG505 SOSIP (this study uses BG505 SOSIP). Do the authors expect any genotypic differences, for example if they repeated the current experiments with B41? Especially since the open state with the occluded FP in the new pocket was in the context of B41. Perhaps that is not easy to test since the B41 sequence lacks the necessary VRC34.01 epitope. But in general, for VRC34.01 sensitive viruses, are the FP transitions and intermediates expected to be similar across different genotypes?

Response:

We have now added a sentence in the discussion: *“Additional studies will also be needed to address the generality of the observed intermediates across different HIV-1 isolates.”*

We would like to note, however, that the generality of the Population 1 structure has now been demonstrated across different studies, in different isolates and in complex with different antibodies. This is described in lines 411-418 (557-574 with track changes on): *“Collective evidence in our study performed using the SOSIP-stabilized Env ectodomain from the BG505 isolate suggests that the structure of Population 1 represents a general intermediate on the HIV-1 entry pathway and that this intermediate is accessible for binding to broadly neutralizing antibodies such as VRC34.01 and 8ANC195. The evidence includes the recurrence of similar geometry in structures obtained from different isolates (BG505 or B41) and in complex with different gp120/gp41 targeting antibodies (8ANC195 and VRC34.01), and their concurrence with the cryo-ET resolved structure of the CD4-bound HIV-1ADA.CM Env in the membrane context.”*

4) Lines 198-201 and Figures C and D: this finding, even if anticipated, is very interesting and really supports the importance of this manuscript. Well done.

Response:

We appreciate the reviewer’s appreciation of the structural visualization of the sequestered FP that provides the explanation for the lack of VRC34.01 binding at the unoccupied sites - *“Examination of these unbound sites revealed FP sequestered within a gp120/gp41 pocket in an antibody-inaccessible configuration (**Figure 3C and D**), thus providing a structural explanation for the lack of antibody binding to these FP sites.”*

5) Figure 3E,F: these measurements are great and help the reader follow the text. In F, I have trouble identifying which color distance relates to which two points (the magenta and brown font in the V1-V2 base distances, and the teal and periwinkle fonts in the CD4bs distances). Perhaps the visualization triangles are accidentally omitted in the uploaded figure (the red and blue overlaid triangles that appear in 4/6 panels). something similar would be helpful here but where each line of the triangles is the color in the distance lists.

Response:

We have now updated Figure 3F to add in the missing triangle, and as the reviewer advised, the color of the lines of the triangles that are overlaid on the structures now match the font color of their distances.

6) Lines 242-253 and figures 4C: it's also possible, related to my earlier comment, that the extended loop versus helix and the size of the pocket could be related to genotypic differences between B41 and BG505. There is a small difference in the FP sequence at HXB2 positions

518-519 in BG505 (VF) and B41 (FI). Perhaps the authors can check and comment whether the different positions of the aromatic side chains have any impact on how the FP arranges in the pocket. Maybe not, if they also saw a similar pocket in population 2 of the BG505 data.

Response:

The difference in the size of the pocket between the 6CM3 (BG505 SOSIP bound to CD4, 17b and 8ANC195; PMID: 30308160) and 5VN3 (B41 SOSIP bound to CD4 and 17b; PMID: 28700571) structures is related primarily to Env conformation, specifically to its extent of opening. The 5VN3 structure is more open creating the larger gp120/gp41 interfacial pocket to stably bury the FP. The 6EDU structure (B41 SOSIP bound to CD4, 17b and 8ANC195; PMID: 30308160) is equivalent to the 6CM3 structure, thus this conformation has been recapitulated across 2 isolates. Moreover, as we have discussed in the manuscript, the 6CM3/6EDU geometry is recapitulated in the Population 1 structure (BG505 SOSIP bound to CD4, 17b and VRC34.01; PDB: 9D90), as well as in the cryo-ET CD4-bound HIV-1ADA.CM Env (EMD-29294; PMID: 32601441). To summarize, the generality of the Population 1 subunit geometry has been established across three HIV-1 isolates, BG505, B41 and ADA.CM.

The larger pocket in the 5VN3 structure occurs upon further Env opening. Similar pockets are observed in the Population 2 and 3 structures in the protomers where the FP is buried (Figures 3C and D), although these pockets are a little smaller than in 5VN3 (Figure 4C, two rightmost panels), suggesting either isolate related differences or that the Population 2 and 3 structures have not opened as completely as the 5VN3 structures. The FP takes an extended conformation within the 5VN3 pocket, whereas in the Population 2 and 3 structures, the FP appears helical. This could also be isolate-related, or it could be related to conformation, i.e., once the pocket in BG505 expands further, the FP can assume the extended conformation.

As stated above, we have added a sentence in the discussion: *“Additional studies will also be needed to address the generality of the observed intermediates across different HIV-1 isolates.”*

7) Table S1: value for "FSC threshold" should be 0.143 instead of 0.5 to match what is reported in the wwPDB validation reports.

Response:

Table S1 now correctly lists a FSC threshold of 0.143 for reporting map resolution.

8) Validation reports: the authors decided to model the constant domains of the Fabs, presumably because the docked in full Fab x-ray structures as the initial model. However, in the map-model overlay and other reported stats in the validation reports, it appears that there is little density or disordered density to support these regions at the recommended threshold. I recommend leaving just the Fv portions in the models as this better agrees with the maps, and should also positively impact the atom inclusion values. Specifically, PDB 9D8Y, 9D90. Same with what I believe is the second domain of sCD4 in PDB 9D98.

Response:

The Fab constant domains and the second domain of sCD4 were deleted and the new coordinates were uploaded on RCSB. Table S1 was updated accordingly.

9) Table S1 and validation reports: minor point but the values in Table 1 are sometimes inconsistent with those in section 4 Experimental information of the validation reports (or section 1 of map only depositions). For example, EMD-46670. Table says 306,507 particles went into the final reconstruction, report lists 259,213. Table says 59.1 electron exposure, report lists 60.9. Please check the other depositions too as each reports a different dose yet Table 1 has the same value for all columns.

Response:

We thank the reviewer for pointing out these inconsistencies between the validation reports and Table S1. These have now been corrected. For the Population 2 structure (EMD-46671), the particle number and the resolution updates have been communicated to the RCSB and will be updated.

10) Considering that VRC34.01 can remain bound for a long time, even in open intermediate states, do the authors think that antibody engineering of VRC34.01, to increase affinity (if possible) would increase the durability? Presumably, VRC34.01 can counter the effects of CD4/17b binding up to a certain threshold.

We thank the reviewer for this interesting suggestion. We performed SPR assays side-by-side with VRC34.01 and its engineered higher affinity variant VRC34.01_mm28 (PMID: 37989731). Under the conditions of our experiment, VRC34.01 and VRC34.01_mm28 were similar in their ability to bind the CD4-activated Env (Figure R3). Since we did not observe much difference between the two antibodies, we have not included this data in the manuscript but have provided the results here.

Figure R3. Time-dependent changes in binding of VRC34.01, VRC34.01 mm28 and 17b to HIV-1 BG505 SOSIP Env upon incubation with CD4. Simultaneous Env opening and fusion peptide burial were measured by incubating Env with CD4 at 37 °C and at different time-points injecting over a VRC34.01 IgG (square datapoints and blue line), VRC3401 mm28 IgG (triangle datapoints and blue line) or a 17b IgG (circle datapoints and orange line) surface.

Response to Reviewers' Comments:

Reviewer #1:

Reviewer #1 (Remarks to the Author):

The authors have addressed most of my comments. However, in the revised version of their paper, the authors present very different binding data in Figures 1c and 1d compared with the data they presented in their initial submission. In the rebuttal letter, they only state that they modified the figure legend, not the figure itself. The authors do not indicate that they performed a different experiment at room temp., raising questions regarding this data and the reason it was replaced (they do indicate a new experiment at 37C incubation, but this is included in Figure S9).

Which graphs faithfully represent the results of these experiments, the graphs that were included in the initial submission or the graphs in the revised version? If both are, why do these graphs look so different, and why didn't the authors explicitly state that they have changed the data shown? The original graph indicated a bi-phasic behavior (which I asked about, but the authors chose not to relate to it in the rebuttal letter), while the revised graph is clearly not bi-phasic. The authors need to explain this discrepancy, how and why the results have changed, and if these are indeed different experiments, why the graphs look different/the results are not reproducible enough.

Response:

Both experiments (initial submission and the revised version) were performed at 25 °C. The revised graphs are a result of more rigorously performed and better controlled experiments. In both experiments (initial submission and the revised version), the components (Env, CD4, 17b) were pre-incubated at 25 °C in a tabletop incubator before mixing. In the experiments shown in the initial submission, post-mixing incubation of the sample was continued in the tabletop incubator with aliquots taken periodically for SPR. The first few injections were performed within a single SPR experiment, while the remaining injections were performed in separate SPR experiments with the sensor chip being docked and undocked, and a new run started each time. In contrast, for the revised graphs, post CD4 (or CD4/17b) addition, the sample was rapidly transferred to the SPR sample chamber pre-equilibrated at 25 °C, was kept within the sample chamber for the duration of the experiment, and the entire experiment was performed within a single SPR run (with low flow-rate injections interspersing the data points). Carrying out the experiment in this manner reduced experimental variability and yielded more consistent and reproducible results.

The new experiments at 37 °C were carried out similarly. The samples were preincubated at 37 °C in a tabletop incubator before mixing. After mixing Env with CD4 (or CD4/17b), the sample was rapidly transferred to the SPR sample chamber, which was pre-warmed to 37 °C. The experiment was carried out within a single SPR run with the assay temperature set at 37 °C and the sample chamber maintained at 37 °C throughout.

Not explicitly mentioning that we changed the graphs for Figures 1C and 1D in our response to reviewer comments was an oversight and we are glad to have the opportunity to clarify here.

The apparent bi-phasic nature of the previous graph and the absence of this in the revised graph was possibly due to the differences in experimental handling. We have not observed any

evidence of bi-phasic behavior in multiple subsequent repeats of the experiments with different batches of protein when the careful temperature control and rapid sample transfer was implemented, and the data for the graph was generated within a single continuous SPR run.

We have updated the SPR methods section as follows to add details of the sample incubation:

Surface Plasmon Resonance

Surface Plasmon Resonance binding assays were performed on a T-200 Biacore system (GE-Healthcare) operating at either 25 °C or 37 °C. HBS-EP+ (10 mM HEPES, pH 7.4, 150 mM NaCl, 3 mM EDTA and 0.05% surfactant P-20) was used as running buffer. A 40 nM solution of BG505.SOSIP prepared in the running buffer was incubated at 25 °C or at 37 °C with either 200 nM of CD4, or with 200 nM CD4 and 200 nM 17b Fab. All samples were pre-incubated at the indicated temperature before mixing. After mixing, the samples were rapidly transferred to the Biacore T-200 temperature-controlled sample chamber which was pre-warmed to 25 °C or 37 °C. The samples were stored within the temperature-controlled sample chamber for the duration of the experiment. The binding surface was prepared by flowing 100 nM each of, 17b IgG and VRC34.01 IgG over each flow cells 2 and 4, respectively at 10 µl/min flow rate for 30 seconds with the 1st and 3rd flow cells serving as reference for 2nd and 4th flow cells, respectively. After surface preparation, the analyte (either BG505 SOSIP alone or BG505 SOSIP with CD4 or BG505 SOSIP with CD4 and 17b Fab) was flowed at 30 µl/min flow rate for 60 seconds. The same injections were carried out using HBS-EP+ buffer to obtain a reference curve. The sensorgrams were blank corrected in the Biacore T-200 evaluation software.

Also, the EM data, in principle, relates to the original protein sample that showed a bi-phasic binding graph. If the protein batch was replaced for new experiments, is there a possibility that the different states would be populated differently?

Unfortunately, these are significant new concerns that the authors will need to address.

Response:

The samples for cryo-EM were prepared by pre-incubating the samples at the desired temperature (25 °C or 37 °C) in a tabletop incubator, mixing Env with CD4 or CD4/17b and returning to the incubator. Due to the different sample concentration requirements, the incubations for SPR and cryo-EM were performed independent of each other. For each cryo-EM dataset a separate sample was prepared and incubated, an aliquot withdrawn at the indicated time and vitrified for cryo-EM analysis.

We have not found any notable variation between protein lots or across independently performed experiments that would alter our results or our interpretations.

Minor points:

The authors need to indicate the map levels in their “zoom-in” views.

Response:

We have now indicated map levels from ChimeraX in the legends for figures where zoomed-in views are shown (Figure 3, S3 and S6).

Also, they should present mesh and not surfaces (as in Figure S6).

Response:

We have now added panels in Figure S6 showing the zoomed-in views with maps in mesh representation. We have edited the Figure legend accordingly.

Reviewer #2:

This manuscript by Thakur et al. is a resubmission. The authors describe conformational states of HIV Env stabilized when bound to VRC34.01, a HIV broadly neutralizing antibody specific for the fusion peptide. Notably, they show FP transitions from a fully VRC34 accessible state in a pre-closed Env trimer, to a FP inaccessible state by changes that occur in the gp120 subunit upon binding to the host CD4 receptor. Soluble trimeric BG505 SOSIP-stabilized gp140 ectodomains were co-incubated with CD4, an antibody that stabilizes a receptor-activated beta-sheet-rich surface on gp120, 17b, and VRC34. Early kinetic intermediate bound states in various stoichiometries were examined using cryoEM, structures were binned and determined. They show how gp120 partially rotates out and opens, upon CD4 engagement, which causes FP to bury in a newly formed gp120-gp41 interfacial pocket.

Response:

We thank the reviewer for this summary of our revised manuscript.

The authors were generally responsive to requests by reviewers and included clarifications requested by this reviewer about how to keep track of and distinguish the multiple substructures being considered. Aside from a minor comment below, this reviewer has no further concerns regarding this manuscript.

Response:

We are glad that the reviewer found our revisions and responses satisfactory and thank the reviewer for their suggestions that helped us improve and clarify the manuscript.

Fig.5C. The text of the descriptions at the bottom of each panel in this figure are too small to read. The text should be enlarged for greater legibility.

Response:

We have enlarged the text at the bottom of each panel in Fig. 5C.

Reviewer #3 (Remarks to the Author):

I am satisfied with the authors' responses to my comments, and for taking the time to edit the manuscript, reprocess some of the data and improve figures. I have no further comments.

Response:

We thank the reviewer for their valuable critiques. We are glad we could address these to the reviewer's satisfaction.

Response to Reviewers' Comments:

Reviewer #1:

Please present all the data for the multiple repeats of the SPR experiments. This could be as additional curves in Figure 1, or as supplementary figures. Also, please state in the figure legend how many times this experiment was repeated.

Otherwise, the authors have addressed all of my comments.

Response:

We now show in Supplementary Figure 1, panel B, two independent repeats of the experiment performed at 37 °C. The data labeled Repeat 1 was used to plot the graph shown in Supplementary Figure 9, panel A. This is clarified in the figure legend: "The dataset labeled Repeat 1 in Supplementary Figure 1, panel B was used for plotting the graph."

We have mentioned in the legend for Figure 1 and Supplementary Figure 1 that "Data shown are representative of at least two independent experiments."

We have further clarified in the legend of Supplementary Figure 1 that "Two independent repeats are shown in panel B, performed at two levels of immobilization. Results were equivalent, showing progressive increase in 17b binding and reduction of VRC34.01 binding post CD4 addition to Env (an effect that was enhanced when 17b Fab was added together with CD4)."

We thank the reviewer for their valuable critiques and are glad that we could satisfactorily address all comments.